# Predictors of elevational biodiversity gradients change from single taxa to the multi-taxa community level

Marcell K. Peters[1], Andreas Hemp[2], Tim Appelhans[3], Christina Behler[4], Alice Classen[1], Florian Detsch[3], Andreas Ensslin[5], Stefan W. Ferger[6], Sara B. Frederiksen[7,8], Friederike Gebert[1], Michael Haas[7], Maria Helbig-Bonitz[4], Claudia Hemp[1], William J. Kindeketa[1,9], Ephraim Mwangomo[3,10], Christine Ngereza[7,11], Insa Otte[3], Juliane Röder[7], Gemma Rutten[5], David Schellenberger Costa[12], Joseph Tardanico[1], Giulia Zancolli[1,13], Jürgen Deckert[14], Connal D. Eardley[15,16], Ralph S. Peters[17], Mark-Oliver Rödel[14], Matthias Schleuning[6], Axel Ssymank[18], Victor Kakengi[19], Jie Zhang[1], Katrin Böhning-Gaese[6,20], Roland Brandl[7], Elisabeth K.V. Kalko[4,21,‡], Michael Kleyer[12], Thomas Nauss[3], Marco Tschapka[4,21], Markus Fischer[5,6] & Ingolf Steffan-Dewenter[1]

The factors determining gradients of biodiversity are a fundamental yet unresolved topic in ecology. While diversity gradients have been analysed for numerous single taxa, progress towards general explanatory models has been hampered by limitations in the phylogenetic coverage of past studies. By parallel sampling of 25 major plant and animal taxa along a 3.7 km elevational gradient on Mt. Kilimanjaro, we quantify cross-taxon consensus in diversity gradients and evaluate predictors of diversity from single taxa to a multi-taxa community level. While single taxa show complex distribution patterns and respond to different environmental factors, scaling up diversity to the community level leads to an unambiguous support for temperature as the main predictor of species richness in both plants and animals. Our findings illuminate the influence of taxonomic coverage for models of diversity gradients and point to the importance of temperature for diversification and species coexistence in plant and animal communities.

[1] Department of Animal Ecology and Tropical Biology, Biocenter, University of Würzburg, Am Hubland, Würzburg 97074, Germany. [2] Department of Plant Systematics, University of Bayreuth, Universitätsstraße 30, Bayreuth 95440, Germany. [3] Environmental Informatics, Faculty of Geography, University of Marburg, Deutschhausstraße 12, Marburg 35032, Germany. [4] Institute for Evolutionary Ecology and Conservation Genomics, University of Ulm, Albert-Einstein-Allee 11, Ulm 89069, Germany. [5] Institute of Plant Sciences, University of Bern, Altenbergrain 21, Bern 3013, Switzerland. [6] Senckenberg Biodiversity and Climate Research Centre (BiK-F), Senckenberganlage 25, Frankfurt am Main 60325, Germany. [7] Department of Ecology, Animal Ecology, University of Marburg, Karl-von-Frisch-Straße 8, Marburg 35032, Germany. [8] Zoological Museum, Natural History Museum of Denmark, University of Copenhagen, Universitetsparken 15, Copenhagen DK-2100, Denmark. [9] Tanzania Commission for Science and Technology, Department of Life Sciences, Ally Hassan Mwinyi Road, PO Box 4302, Dar es Salaam, Tanzania. [10] Mount Kilimanjaro National Park, PO Box 96, Marangu, Moshi, Tanzania. [11] National Museum of Tanzania, Shaaban Robert Street, Dar es Salaam, Tanzania. [12] Landscape Ecology Group, Institute of Biology and Environmental Sciences, University Oldenburg, Oldenburg 26111, Germany. [13] Molecular Ecology and Fisheries Genetics Lab, School of Biological Sciences, Environment Centre Wales, Bangor University, Bangor LL57 2UW, UK. [14] Museum für Naturkunde, Leibniz Institute for Evolution and Biodiversity Science, Invalidenstraße 43, Berlin 10115, Germany. [15] Agricultural Research Council—Plant Protection Research: Plant Health and Protection, Private Bag X134, Queenswood, Pretoria 0121, South Africa. [16] School of Life Sciences, University of KwaZulu-Natal, Private Bag X01, Scottsville, Pietermaritzburg 3209, South Africa. [17] Zoological Research Museum Alexander Koenig, Department Arthropoda, Adenauerallee 160, Bonn 53113, Germany. [18] Falkenweg 6, Wachtberg 53343, Germany. [19] Tanzania Wildlife Research Institute, PO Box 661, Arusha, Tanzania. [20] Institute for Ecology, Evolution and Diversity, Goethe University Frankfurt, Biologicum, Max-von-Laue-Straße 13, Frankfurt am Main 60439, Germany. [21] Smithsonian Tropical Research Institute, PO Box 0843-03092, Balboa Ancòn, Republica de Panamà. Correspondence and requests for materials should be addressed to M.K.P. (email: marcell.peters@uni-wuerzburg.de).
‡Deceased 26 September 2011

The search for the primary factors that determine the distribution of biodiversity on earth has challenged naturalists for more than two centuries[1–3]. The main hypotheses for explaining broad-scale diversity gradients are: (1) the 'temperature hypothesis', relating higher species richness to higher rates of biotic processes and interactions, or higher evolutionary diversification rates[4–6]; (2) the 'water availability hypothesis', focusing on direct or indirect (via effects on net primary productivity (NPP)) constraints of water for the maintenance of biodiversity[7,8]; (3) the 'productivity hypothesis', emphasizing the positive effect of resources on population persistence and species coexistence[9–11]; (4) the 'area hypothesis', assuming greater opportunities for the maintenance of richness and speciation in larger areas of land (or sea)[12–14]; (5) the 'geometric constraints hypothesis', focusing on spatial constraints for the distribution of species ranges, predicting geographic gradients in species richness even in the absence of environmental drivers[15,16]; and (6) the 'plant diversity hypothesis', relating consumer richness to the species richness of plant resources[17,18]. While numerous studies on the diversity of individual taxa along different environmental gradients found support for one or the other hypothesis, comparative cross-taxon studies along the same environmental gradients are scarce, impeding consensus on the determinants of broad-scale diversity gradients[2,19].

A persistent shortcoming in published gradient studies is that trends in species richness were mostly studied for organismic groups at a narrow taxonomic level (for example, for taxa such as ferns, ants or birds) but not for the multi-group communities of animals and plants which coexist and share resources in ecosystems. However, as species and clades tend to retain their niches and related ecological traits over evolutionary time (niche conservatism), narrowly defined taxa can be more strongly constrained to specific biotic and abiotic conditions along larger environmental gradients than communities composed of multiple, phylogenetically largely independent taxa[20,21]. Patterns of species richness of a specific taxon may, therefore, be idiosyncratic rather than congruent to patterns in other taxa or to those of the community as a whole. Along large environmental gradients, taxonomically diverse communities have larger opportunities for diversification and the use of niche space than a single taxon. For example, while bee species are typically restricted to warm environments and their species richness peaks in hot climates, syrphid flies have their highest density at considerably cooler temperatures, such that the thermal niche space used by pollinators in total is much larger than the thermal niches of the individual taxa[22,23]. Broadening the taxonomic scope may therefore lead to changes in the importance of some potential drivers of diversity over others. In particular, energy variables like temperature and NPP have been predicted to increase in importance for more broadly covered species communities[6,11,24]. However, to our best knowledge, the influence of taxonomic coverage for the extrapolation of broad-scale drivers of diversity has hitherto not been tested.

We used a novel, multi-taxa perspective to study patterns and drivers of species richness along a 3.7 km elevational gradient (871–4,550 m above sea level (a.s.l.)) of natural habitat on the southern slopes of Mt. Kilimanjaro (Supplementary Fig. 1). Tropical mountains with a dry base, such as Mt. Kilimanjaro, are particularly suitable to test large-scale drivers of biodiversity because, in contrast to most terrestrial latitudinal gradients, their temperature and primary productivity gradients are largely uncorrelated (Supplementary Fig. 2). We assessed species richness of eight vascular plant taxa (that is, all vascular plants), and 16 major animal taxa synchronously on the same study sites. The studied taxa were phylogenetically highly diverse, covered multiple trophic levels and represented significant proportions of diversity found in terrestrial plant and animal communities[25] (Fig. 1a).

We, firstly, analysed congruence in the patterns and predictors of elevational diversity among single taxonomic groups, that is, for taxonomic groups which are typically studied in macroecology. Secondly, we analysed how increasing the taxonomic coverage towards the community level modifies patterns of elevational diversity and the support for hypotheses explaining them. Here we show that disparate factors drive the distributions of individual taxa. However, scaling up diversity to the community level leads to a monotonic decline of species richness with elevation and provides a strong support for temperature as the major predictor of plant and animal species richness. Our study reveals high variation in diversity gradients of narrowly defined taxa and underscores the significance of taxonomically complex data sets for understanding the broad-scale drivers of biodiversity.

## Results

**Elevational species richness of single taxa**. We detected a variety of elevational patterns in species richness among plant and animal taxa (Fig. 1b). Approximately half of the plant and animal taxa exhibited monotonic declines in species richness with increasing elevation (Fig. 1b and Supplementary Figs 3–4). The other half had hump-shaped or more complex (for example, bimodal) distribution patterns. While the plant censuses were complete and the species richness in each study site was accurate, incomplete sampling, a regular problem in studies of tropical communities[18,26], could have had biased patterns of elevational diversity in animals (Supplementary Table 1). Estimates of sample coverage, a measure of sampling completeness[27], of most animal taxa were high (mean sample coverage $> 0.80$, s.d. $< 0.17$) but values in four taxa (in spiders, parasitoid wasps, other aculeate Hymenoptera and moths) were moderate to low (mean sampling coverage: 0.54–0.72, s.d. $= 0.21$–0.38). This suggested that further sampling could have significantly increased estimates of species richness (Supplementary Table 1). As sample coverage was significantly positively correlated with elevation in some animal taxa (Collembola, true bugs, bees, other aculeate Hymenoptera; Pearson's product moment correlation, $r = 0.37$–0.85, $P < 0.05$) we checked if incomplete sampling could have biased elevational diversity patterns. We calculated for all animal taxa estimates of asymptotic species richness with the non-parametric Chao1 index[27,28] and compared diversity trends to those detected in the observed data. Patterns of elevational diversity based on estimated asymptotic species richness were highly similar to those derived from the observed data (mean ± s.d. of Pearson's $r = 0.96 \pm 0.09$, Supplementary Table 2).

In the next step we quantified the support for different hypotheses to explain species richness data at the level of single taxa using information-theory based multi-model averaging[29,30]. As potential predictors of diversity we used temperature (in animals: mean annual temperature, MAT; and in plants: mean minimum temperature, MMT[31]), mean annual precipitation (MAP, water availability hypothesis), net primary productivity (NPP; productivity hypothesis), land area within an elevation belt of 300 m above and below the level of the study sites (area hypothesis), predictions of a mid-domain effect (MDE) resulting from multiple random arrangements of species ranges within the elevation boundaries on Mt. Kilimanjaro (geometric constraints hypothesis) and in case of species richness of animals, plant species richness (PSR, plant diversity hypothesis) (Supplementary Fig. 2).

Species richness was best predicted by sets of variables which differed among the individual taxonomic groups (Table 1 and

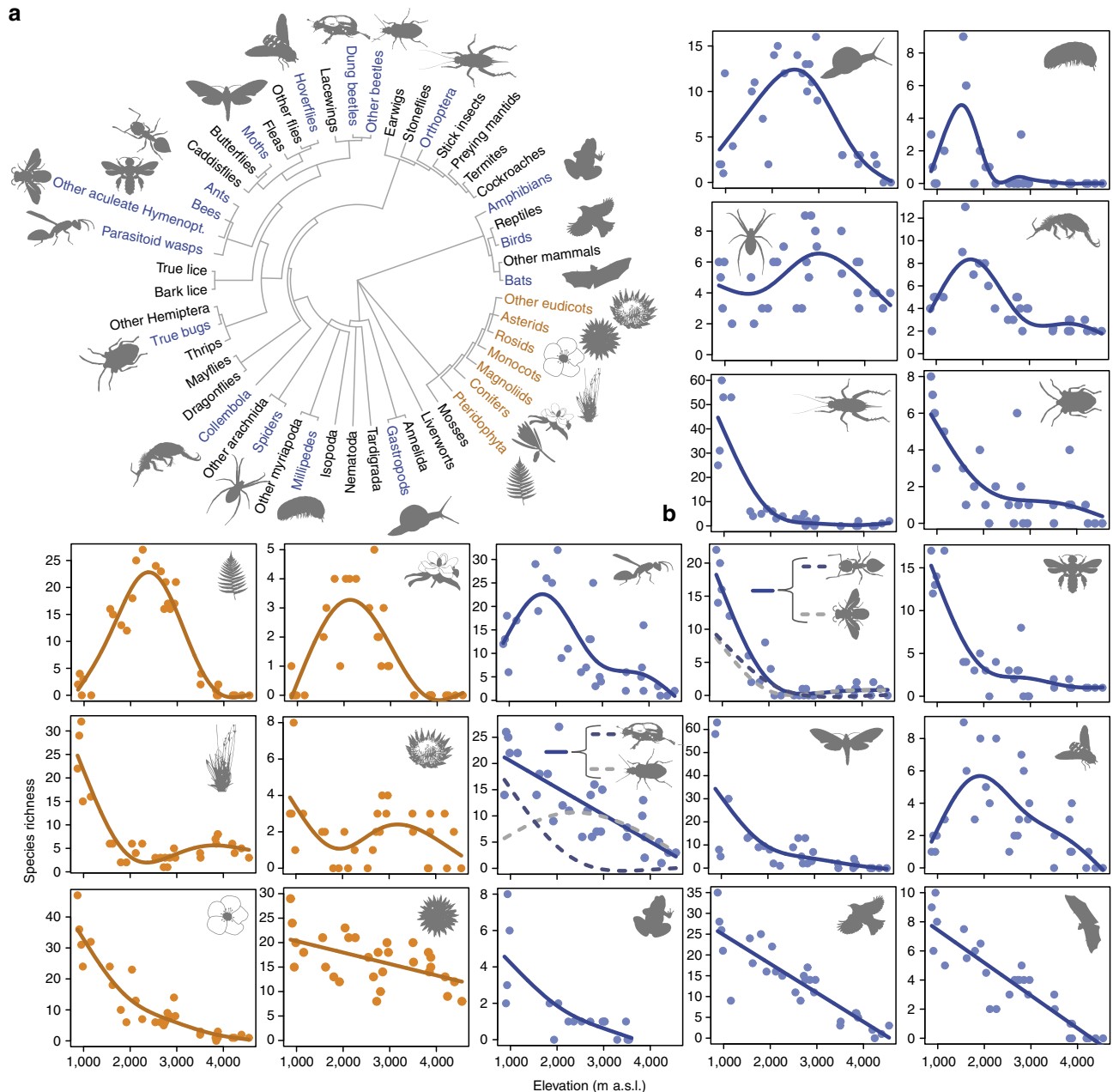

**Figure 1 | Patterns of elevational species richness of single taxa. (a)** Phylogenetic distribution of studied taxonomic groups among major terrestrial plant and animal lineages. **(b)** Patterns of elevational species richness for vascular plants (orange) and animals (blue). Dots represent original measurements on study sites (plants: $N = 30$; animals: $N = 30$, except for Collembola ($N = 29$), ground-dwelling beetles ($N = 29$) and amphibians ($N = 17$)). Trend lines were calculated using generalized additive models; all trends were significant ($P < 0.05$). For trend lines of additional plant groups with low numbers of species see Supplementary Fig. 3. The plant and animal images used in the figure are licensed for use in the Public Domain without copyright, except for the images used for monocots (F. & K. Starr, modified), bats (O. Peles and Y. Wong), Collembola, ground-dwelling beetles (B. Lang), Orthoptera, other aculeate wasps, bees (M. Menchetti), gastropods (G. Monger) that are licenced under a Creative Commons Attribution 3.0 Unported licence (https://creativecommons.org/licenses/by/3.0). Oliver Niehuis kindly provided the permission to use the image of the parasitoid wasp.

Supplementary Table 3). Temperature (MMT and MAT) was an important predictor of species richness in most taxa, having significant positive effects in 11 plant and animal taxa and a negative effect in one taxon. NPP was significantly correlated to species richness in eight plant and animal groups. However, positive and negative effects of primary productivity were equally common. Six taxa exhibited significant positive, and three taxa negative relationships to MAP. Richness of phylogenetically basal plant lineages (Lycopodiopsida, ferns, conifers) tended

to increase, while richness of more modern lineages (for example, monocots) tended to decrease at elevated levels of MAP (Table 1). Significant positive effects of land area and geometric constraints (that is, the MDE) were only observed in two taxa (millipedes and Orthoptera). In moths and gastropods, two animal groups with completely or largely herbivorous diet, significant, positive correlations with PSR were detected. Other animal taxa did not strongly respond to the variation in plant richness.

**Table 1 | Synthesis models explaining species richness of plant (a) and animal taxa (b) derived by multi-model averaging.**

| | | | Conditional standardized estimates†† | | | | |
|---|---|---|---|---|---|---|---|
| Taxon | #species* | #models† | MMT | NPP | MAP | Area | MDE |
| Ferns | 64 | 11 | 0.63 | 0.32 | 0.40 | −0.62 | 0.42 |
| Magnoliids | 9 | 8 | 0.34 | 0.21 | 0.72 | −0.28 | 0.11 |
| Monocots | 106 | 6 | 0.92 | −0.55 | −0.40 | −0.11 | −0.28 |
| Other eudicots | 26 | 4 | 0.15 | 0.80 | −0.95 | 0.08 | −0.19 |
| Rosids | 158 | 8 | 0.88 | −0.28 | −0.18 | 0.03 | −0.31 |
| Asterids | 189 | 6 | 0.55 | −0.05 | −0.09 | 0.16 | −0.05 |
| All vascular plants | 557 | 10 | 1.08 | −0.22 | −0.17 | −0.42 | −0.19 |

| | | | Conditional standardized estimates†† | | | | | |
|---|---|---|---|---|---|---|---|---|
| Taxon | #species* | #models† | MAT | NPP | MAP | Area | MDE | PSR |
| Gastropods | 45 | 7 | 0.23 | 0.01 | 0.03 | 0.27 | 0.77 | 0.39 |
| Millipedes | 14 | 6 | −1.37 | 0.57 | 0.57 | 1.53 | −0.23 | 0.31 |
| Spiders | 52 | 30 | 0.01 | 0.59 | −0.51 | −0.73 | 0.34 | 0.56 |
| Collembola | 23 | 11 | −1.03 | 1.04 | 0.48 | 1.03 | −0.75 | −0.44 |
| Orthoptera | 114 | 9 | 0.57 | −0.05 | −0.30 | 0.74 | 0.01 | −0.13 |
| True bugs | 44 | 16 | 0.81 | −0.20 | −0.19 | −0.21 | −0.20 | ~0.00 |
| Parasitoid wasps | 223 | 13 | 0.36 | 0.45 | 0.50 | 0.47 | 0.03 | −0.19 |
| Ground–dwelling ants | 38 | 20 | 1.00 | −0.33 | −0.21 | 0.47 | −0.17 | −0.24 |
| Bees | 52 | 6 | 1.08 | −0.42 | −0.09 | −0.05 | 0.18 | −0.11 |
| Other aculeate Hym. | 45 | 12 | 1.13 | −0.57 | −0.14 | −0.56 | −0.04 | 0.27 |
| Dung beetles | 56 | 18 | 1.09 | −0.28 | −0.22 | 0.62 | −0.01 | −0.36 |
| Ground–dwelling beetles | 122 | 27 | −0.41 | 0.58 | 0.04 | −0.25 | 0.34 | 0.37 |
| Moths | 199 | 16 | 1.08 | 0.26 | −0.02 | −0.79 | −0.38 | 0.54 |
| Hoverflies | 19 | 13 | −0.02 | 0.04 | 0.63 | 0.37 | 0.34 | 0.02 |
| Birds | 122 | 7 | 0.87 | 0.78 | 0.07 | −0.66 | −0.68 | 0.29 |
| Aerial insectivorous bats | 16 | 15 | 0.93 | 0.27 | −0.13 | −0.25 | ~0.00 | −0.22 |
| All animals | 1184 | 6 | 0.96 | −0.01 | −0.02 | 0.02 | −0.03 | 0.06 |

Shown are standardized parameter estimates for all predictor variables derived from weighted averaging of parameter estimates over best-fit models. Colours indicate significant ($P < 0.05$) positive (blue) or negative (red) effects from multi-model averaging analyses. Results for two additional plant groups with low numbers of species are presented in Supplementary Table 3.
*Total number of detected species/morphospecies for each taxon.
†Number of best-fit models ($\Delta$AIC < 4) used for inference on parameter estimates and variable importance.
††Standardized estimates (standardized beta) over all best-fit models including the respective predictor variable.
Other aculeate Hym., other aculeate hymenoptera; MAT, mean annual temperature; MAP, mean annual precipitation; MDE, mid-domain effect prediction; MMT, mean minimum temperature; NPP, net primary productivity; PSR, plant species richness.

**Elevational species richness of multi-taxa communities.** Enhancing the taxonomic coverage of the data from one to eight (in plants) or 16 taxa (in animals) increased the explained variation of statistical models (that is, of species richness as a function of elevation) and lead to an increasing linearization of elevational species richness patterns in both plants and animals (Fig. 2). At the full multi-taxa community level (that is, species richness calculated by summing species richness values of all studied plant or animal taxa), species richness declined linearly with elevation in plants and quasi-linearly in animals (Fig. 3) with high levels of explained variation (generalized additive models; plants: explained deviance = 77%, estimated degrees of freedom (edf) = 1, $F = 94.4$, $P < 0.001$; animals: explained deviance = 92%, edf = 2.34, $F = 105$, $P < 0.001$). The decline of biodiversity with elevation was also evident but less pronounced when analysing richness of higher taxonomic levels (Fig. 3b,d): the number of orders and families (the latter only in plants) remained relatively constant up to mid-elevations and strongly declined at higher elevations.

Increasing the taxonomic coverage of the data from single taxa to the multi-taxa community level revealed an increasing importance of MMT and MAT as the main predictors of diversity, while the support for NPP, area and other predictor variables declined (Fig. 4). At the highest level of taxonomic coverage, MMT and MAT were the only significant predictors of species richness (Table 1), explaining 79 and 94% of the total variation in plant and animal species richness, respectively. Community level results were highly robust against the exclusion of single or multiple taxa (Fig. 4) and to differences in sampling intensity among taxa (Supplementary Fig. 5).

To document the relationships among predictor variables and to disentangle direct from indirect effects we calculated path models (Fig. 5). As suggested by analyses along other broad-scale gradients[32], precipitation and temperature were strongly linked to the NPP of ecosystems. The direct effects of temperature on PSR were much stronger than the indirect effects via temperatures' positive influence on NPP. Similarly, in path analyses including animal species richness as the final endogenous variable, the direct effect of temperature on animal richness was much stronger than the indirect effects mediated via its positive influences on NPP and PSR.

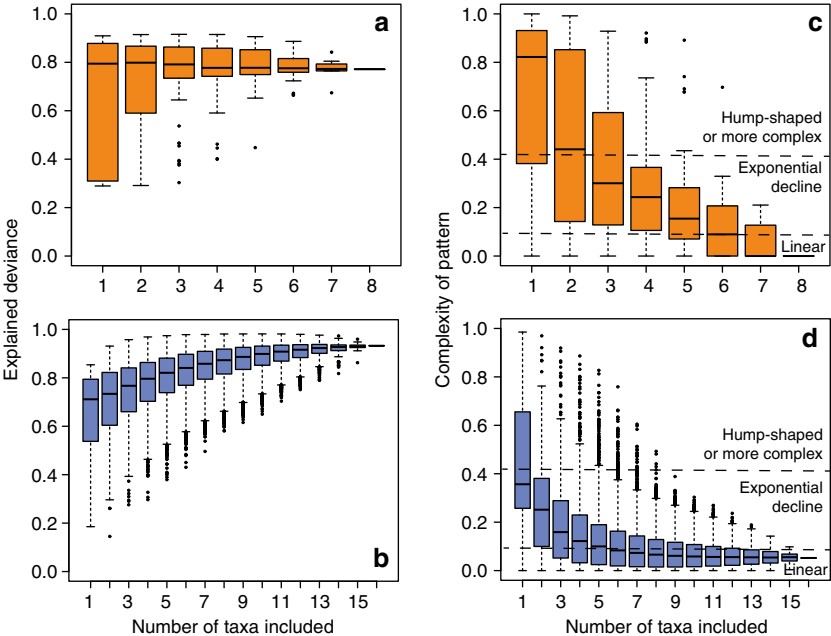

**Figure 2 | Elevational species richness patterns with increasing taxonomic coverage.** (**a,b**) The explained deviance of generalized additive models increased with increasing taxonomic coverage of plant (orange: **a,c**) and animal (blue: **b,d**) communities. (**c,d**) While single taxonomic groups showed a variation of elevational species richness patterns (that is, linear decline, exponential decline or hump-shaped distributions) increasing the taxonomic coverage unambiguously led to patterns of linear decline in both plants and animals. In individual box-and-whisker-plots, bold lines indicate the median, boxes the interquartile range. Whiskers extend to the maximum and minimum values but end at 1.5 × the interquartile range. More extreme data are plotted as single dots.

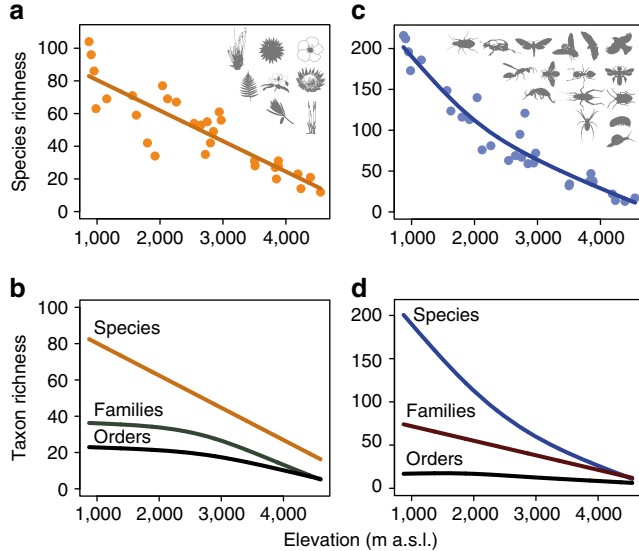

**Figure 3 | Elevational species richness at the community level.** Species richness of vascular plants (**a**) and animals (**c**) along the elevational gradient of Mt. Kilimanjaro. Lower panels show trend lines for the number of species, families and orders of vascular plants (**b**) and animals (**d**) along the elevational gradient. All trend lines were calculated using generalized additive models ($N = 30$ and $N = 29$ in plants and animals, respectively; all trends were significant at $P < 0.001$). Please see Fig. 1 for credits to the authors of the original plant and animal images.

## Discussion

Despite decades of ecological research there is little consensus concerning the determinants of diversity gradients[13,19]. Our study of species richness patterns of multiple plant and animal taxa along the same environmental gradient on Mt. Kilimanjaro,

Tanzania, revealed that there is no general model for explaining diversity gradients across separately analysed taxonomic groups. Instead, patterns and predictors of elevational diversity appear to be idiosyncratic rather than uniform among taxa as they depend on taxon-specific resource requirements and adaptations to the environment. However, we found that the importance of potential drivers of diversity depends on the taxonomic coverage of analyses with taxonomic upscaling unambiguously revealing temperature as the single, well-supported predictor of species richness for taxonomically broad communities. Our results suggest multi-taxa community diversity as a new approach to develop more general models to explain trends in biodiversity: while past studies focused on congruence across different taxonomic groups, a predictor may be seen as having higher generality if it explains trends in diversity for taxonomically more inclusive species communities. Even though former studies already assessed biodiversity of several plant or animal taxa along the same elevational gradient[33–36], our study is unprecedented in terms of taxonomic coverage allowing generalizations and conclusions which could not be achieved with former data sets.

A monotonic decrease of species richness with elevation, as found at the highest level of taxonomic coverage, appears to contrast with former meta-analyses[37,38] finding unimodal patterns of elevational diversity in a majority of studies. However, all of the individual studies on animals and many of those on plants reanalysed in meta-analyses are based on taxonomically restricted data sets, that is, these meta-analyses are based on taxonomic groups as used in our single taxa analyses. Analyses at this taxonomic level revealed also for the Kilimanjaro region a high percentage of unimodal patterns (in 44% of all tested taxa diversity patterns were unimodal). However, with increasing taxonomic coverage of plant and animal communities, unimodal patterns disappeared and the elevational diversity patterns shifted towards a monotonic decline of richness. This unification of diversity patterns with increasing

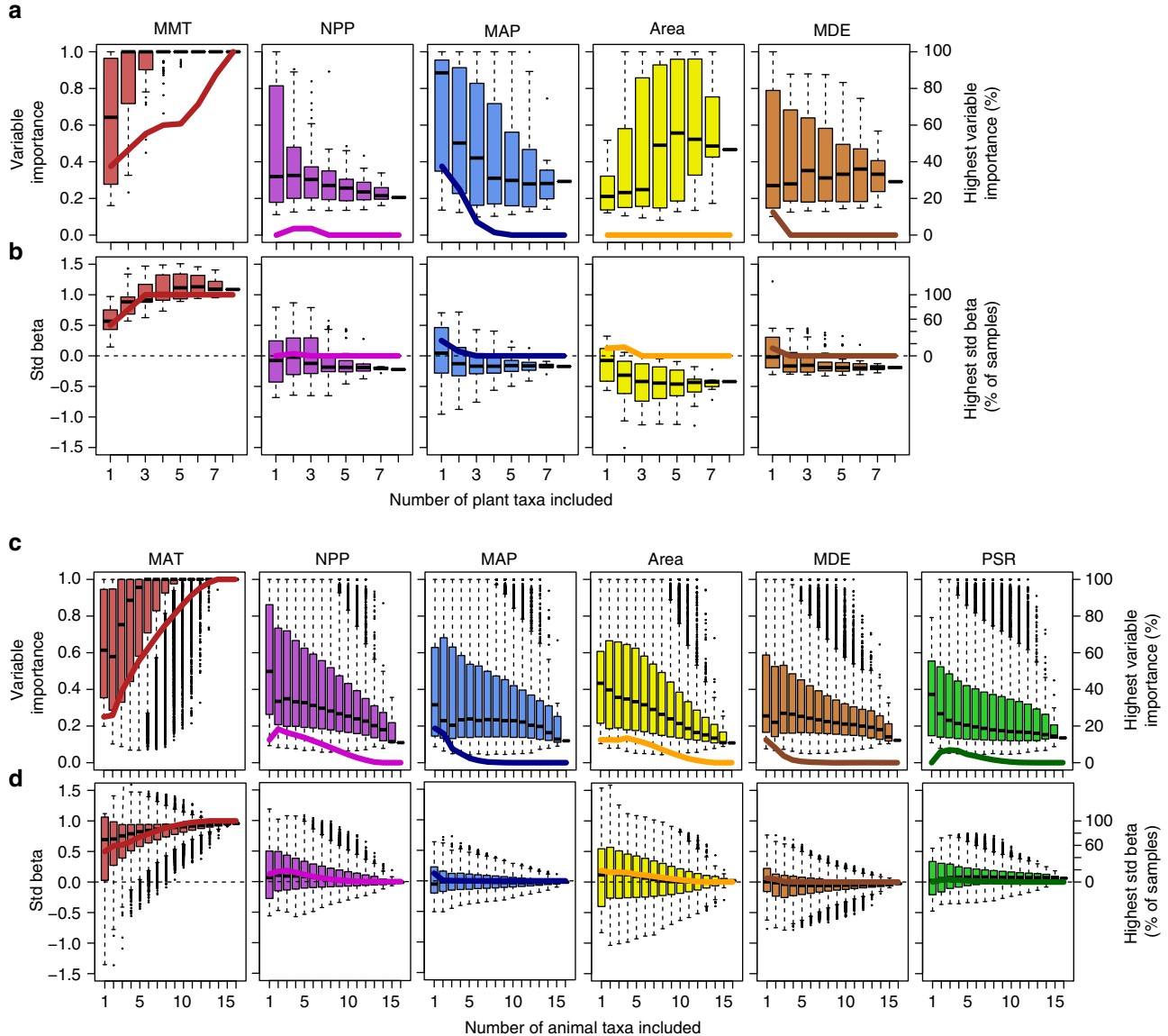

**Figure 4 | Statistical support for predictors of species richness in relationship to taxonomic coverage.** Box-and-whisker-plots show the variation in measures of variable importance (**a,c**) and standardized beta (**b,d**) in relationship to the taxonomic coverage of plant (**a,b**) and animal communities (**c,d**) (values correspond to the left y axis). Bold curved lines give the percentage of all possible taxa combinations in which a variable had the highest variable importance or standardized beta value (extending from 0 to 100%, right y axis). Variable importance is defined as the sum of the Akaike weights of all best-fit models which include the respective predictor variable. Standardized beta values are standardized parameter estimates derived from conditional weighted averaging of parameter estimates over best-fit models. MMT, mean minimum temperature; MAT, mean annual temperature; NPP, net primary productivity; MAP, mean annual precipitation; MDE, mid-domain effect prediction; PSR, plant species richness.

taxonomic scale was also observed within some clades, for example, in beetles and Hymenoptera. Future studies on other mountains and along other environmental gradients will be valuable to proof the generality of our findings in other biogeographic and climatic regions.

Taxonomic groups like birds, ants or terrestrial gastropods strongly differed in their elevational diversity distributions and their species numbers were best predicted by different sets of predictor variables. In approximately 50% of the taxa temperature was the strongest predictor of species richness, but for the other half other variables appeared to be of higher importance. This idiosyncratic response of taxa to environmental gradients is best explained by niche conservatism, that is, the tendency of lineages to retain their niches and related ecological traits over evolutionary time[20,39–41], leading to phylogenetic autocorre-

lation in elevational distributions (Supplementary Fig. 6) and clade-specific species distributions patterns[42,43] (Fig. 1b). Strong constraints in the evolution of niche space (and of related traits and resource use) may result in a low fit of diversity to energy variables in narrowly defined taxa[6,11]. However, the fit may increase when extending the taxonomic coverage of communities (Table 1 and Fig. 4).

The integration of multiple taxonomic groups in analyses unequivocally revealed that community level diversity is mainly predicted by temperature. The increasing importance of temperature on diversity for taxonomically more broadly covered communities was driven by two effects: Even though temperature was neither the most important nor a consistently significant predictor variable across plant and animal taxa, its effect was positive, reaching from subtle to strong, in the majority of taxa

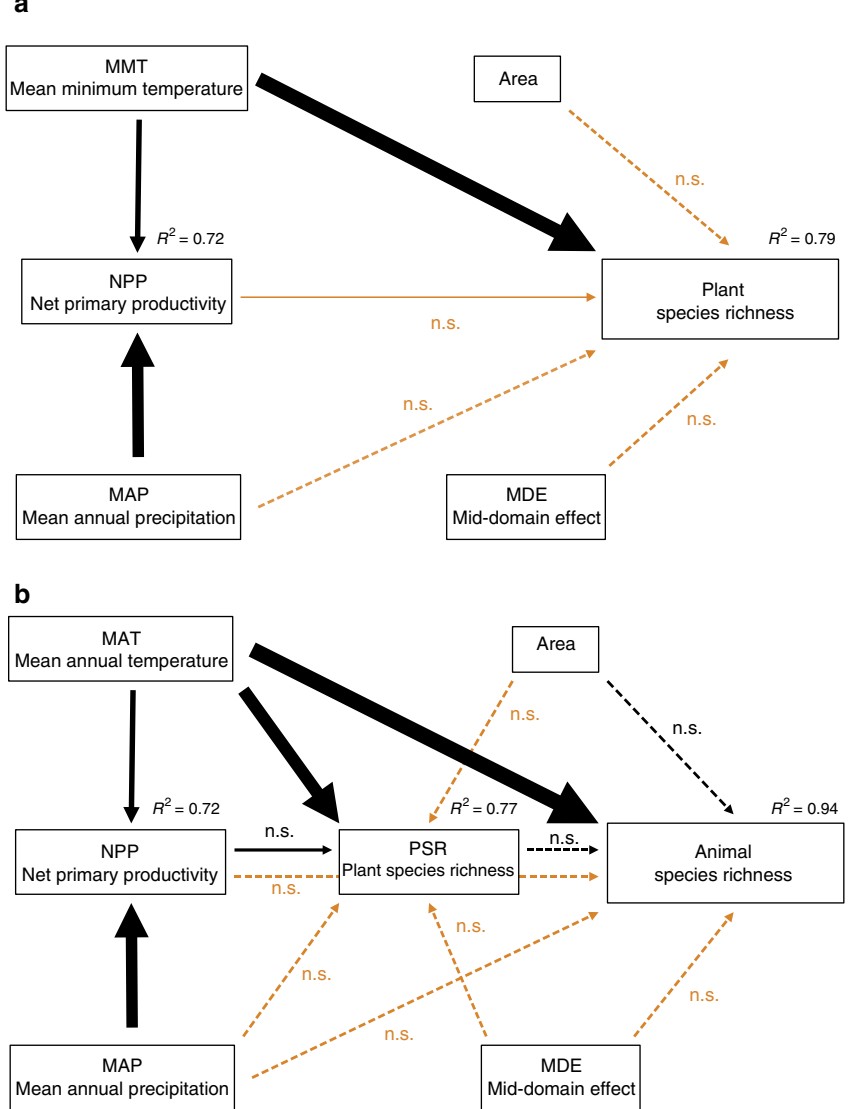

**Figure 5 | Path models showing direct and indirect effects of predictor variables on species richness.** For plants (**a**) and animals (**b**), the path model with the lowest Akaike information criterion (AICc) is presented as solid lines. Interrupted lines indicate potential paths used for the construction of competing models (all models with ΔAICc < 3 identified by multi-model inference) but which were excluded from the final path model. For all paths of the final path model, arrow width is proportional to the relative strength of standardized path coefficients. Orange and black arrows indicate negative and positive effects, respectively. For each endogenous variable the relative amount of explained variance is given. n.s., not significant.

(Table 1). In addition, many of the taxa exhibiting strong positive temperature–diversity relationships (for example, monocots, rosids, asterids; Orthoptera, all groups of aculeate Hymenoptera, birds, moths) were more diverse, contributing more species to the local species communities, than taxa with negative or weak relationships (for example, magnoliids, millipedes, Collembola; but note that there were also exceptions, for example, the diverse group of ground-dwelling beetles). The largely positive effect of temperature on species richness (Table 1) was independent of differences in sampling intensity among taxa (Supplementary Fig. 5) and contrasts with the heterogeneous effects of other predictor variables: NPP was a well-supported predictor variable only for some single taxa but not for the multi-taxa community[44]. This was because positive and negative effects of NPP on species richness were equally common in single-taxa analyses. Species richness of most taxonomic groups and of the animal and plant communities as a whole was highest in the dry savannah where primary productivity and precipitation is rather low, suggesting

that even low levels of primary productivity and temporal aridity do not principally put major limitations to the diversification or species coexistence of plant or animal assemblages as a whole. High levels of statistical support for the MDE, which is often used to explain unimodal patterns of elevational diversity[15,16], was rare for single taxa and the support decreased with increasing taxonomic coverage. The species richness of a large number of taxa peaked at the lowest elevation and, even in taxa exhibiting unimodal distributions of species richness, the highest richness values were not generally observed at the middle of the domain but at variable elevations (Fig. 1 and Supplementary Fig. 4). These observations suggest a limited role of this type of geometric constraints for the elevational distribution of plant or animal diversity on Mt. Kilimanjaro. However, even though the MDE has often been used to explain patterns of diversity on other mountains which do not extend to the sea level[45,46], its application to these systems is debated[15]. Moreover, recent studies proposed to refine the MDE models by including

midpoint attractors, that is, peaks of favourable environmental conditions, in models to improve predictions of diversity gradients[47]. The area available to species populations is regarded as a major determinant of local and regional species richness[12,13]. However, land area was a poor predictor of plant and animal species richness on Mt. Kilimanjaro, where significant positive effects were only found for millipedes and Orthoptera. PSR received high statistical support as a predictor of animal species richness only in terrestrial gastropods and moths. Particularly, moths are well known for their specialization on certain food plants which probably facilitates strong associations between herbivore and plant diversity[48].

The large support for MMT and MAT as the major predictors of multi-taxa species richness suggests a high importance of temperature-driven mechanisms facilitating the origination and maintenance of biodiversity[24,49]. While recent studies emphasize the positive effect of temperature on speciation rates[2,50,51], the young age of Mt. Kilimanjaro ($<2$ Mio years) and the fact that the trends were found for diversity estimates taken at local rather than regional spatial scales (study sites of $50 \times 50$ m) suggest the additional importance of ecological mechanisms, such as positive effects of temperature on rates of negative-density dependence[52,53] or positive effects on resource exploitation and other biological rates[23,49].

While species richness data for many different taxonomic groups were assessed and sampling effort for most taxa was high, several species-rich animal groups were not studied and additional sampling would probably yield more species in local communities. Nonetheless, several points underpin that our results are robust against the inclusion of additional taxonomic groups or increased sampling effort: (i) No taxon considered in this study was included based on an *a priori* expectation of a certain elevational species richness pattern, and the taxa studied were phylogenetically highly diverse and represent much of the terrestrial plant and animal phylogenetic diversity[25] (Fig. 1a). (ii) The sampling resembled a stratified random sampling approach (first strata: sampling of larger taxa, second strata: sampling of species within taxa), which provides comparative estimates of community level diversity even under incomplete sampling of taxa and species (Supplementary Fig. 7). (iii) Patterns for all taxonomic groups remained consistent when controlling for differences in sampling completeness (Supplementary Table 2) and analyses restricted to taxa sampled with high effort did not change resulting patterns (Fig. 4). (iv) Moreover, community-level analyses in which we applied a rarefaction approach to standardize sampling effort across animal taxa lead to the same conclusions regarding the importance of temperature as the major predictor of species richness (Supplementary Fig. 5). (v) Last, analysing only a fraction of the full data set of taxa unambiguously revealed temperature as the most important predictor of plant and animal species richness in terms of both variable importance and effect strength (Fig. 4).

While our study was extensive in terms of studied taxonomic diversity, it was, however, restricted to a single mountain with a specific geological, climatic and biogeographic context. For example, patterns of elevational diversity have been shown to depend on the extent of the elevational gradient[38]. Even though the lowest elevations studied at Mt. Kilimanjaro form the natural base of the mountain (as the Mt. Kilimanjaro region and most parts of East Africa are situated at high elevations), equatorial mountains extending to the sea level could provide even more extreme environmental gradients and could show distinct elevational patterns. In addition, our interpretation of the importance of temperature and other predictor variables is based on hypotheses-driven correlative analyses of species distribution data. It will be highly interesting to evaluate

the support for drivers of multi-taxa diversity with both correlative and experimental approaches on other mountains, differing in age, climate, degree of isolation and biogeographic context.

In conclusion, our study revealed that a broad taxonomic coverage in macroecological studies provides new insights into the drivers of broad-scale diversity gradients. Whereas upscaling from regional to global patterns significantly enhanced the perception of the major drivers of diversity[8,13,44], increasing the phylogenetic coverage of studies may be another axis to consider on the way to generalization and causal understanding. While geometric constraints, area, water and productivity variables were of importance only for predicting richness of few groups of low taxonomic level, temperature gained consistent, increasing support as a main predictor of species richness with increasing taxonomic coverage of communities. Identifying the ecological and evolutionary mechanisms by which temperature governs the distribution of biodiversity will be of fundamental importance for understanding global gradients of diversity and the long-term consequences of global warming.

## Methods

**Time and area of study.** All data were collected from December 2011 through January 2014 on the southern and south-eastern slopes of Mt. Kilimanjaro (Tanzania, East Africa; 2°45′-3°25′S, 37°00′-37°43′E). Mt. Kilimanjaro has a northwest-southeast diameter of ~90 km and rises from the savannah plains at 700 m elevation to a snow-clad summit at 5,895 m a.s.l. Precipitation is bimodal with the main rainy season occurring from March through May and the more variable short rains around November. The MAT decreases in a quasi-linear manner with elevation having an overall lapse rate of ~0.56 °C per 100 m starting with 25 °C at the foothills and decreasing to −8 °C at the top of the mountain[54]. Vertical precipitation distribution shows an unimodal pattern, with 600–900 mm year$^{-1}$ at the base of the mountain, ~2,500–3,000 mm year$^{-1}$ at 2,200 m a.s.l. and <500 mm year$^{-1}$ in the Afroalpine zone at the highest elevations[54,55]. Because of a long history of human impact, natural habitats in the lowlands were largely cleared[54,55]. Habitats above 1,800 m a.s.l. are protected as a national park (Mt. Kilimanjaro National Park).

**Phylogenetic tree.** To demonstrate the distribution of studied taxa across the whole phylogenetic diversity of free-living land plant and animals we constructed a phylogenetic tree of all major plant and animal taxa. Here only the major taxa, that is, taxa with more than 1,000 described species, were considered. For identifying the plant and animal taxa with more than 1,000 described species we used Chapman[25]. We filtered out all predominately aquatic taxa (for example, most animal phyla nearly exclusively occur in marine environments). Taxonomic and phylogenetic data in http://en.wikipedia.org (accessed 25 July 2014), Trautwein et al.[56] and Misof et al.[57] was used to construct a phylogenetic tree of the major terrestrial taxa using the R package ape with branch length computation based on Grafen[58]. Please note that while the topology of the tree well represents current knowledge, branch lengths are, however, arbitrary and not indicative of clade ages or rates of molecular evolution. We consider this approach as sufficient because the tree is solely used for demonstrating the phylogenetic distribution of studied taxa and was not used for any statistical inference.

**Study design.** We established thirty $50 \times 50$ m study sites, which spanned an elevational gradient of 871–4,550 m a.s.l. Lower elevations are scarcely found in the East African highland region, where Mt. Kilimanjaro is located, and the lowest elevations incorporated in our studied elevational gradient can be considered as a typical, natural base of the mountain (within a square of 2° × 2° (~222 × 222 km) with the Kibo peak of Mt. Kilimanjaro in the centre, 100% of the area is >500 m a.s.l., 99% >600 m a.s.l., 94% >700 m a.s.l. and 83% >800 m a.s.l.). The study sites were equally distributed over the six major types of natural habitats found along the south-eastern slope of Mt. Kilimanjaro: savannah woodland (~800–1,150 m a.s.l.), submontane and lower montane 'Newtonia' forest (~1,150–2,050 m a.s.l.), 'Ocotea' forest (~1,800–2,800 m a.s.l.), 'Podocarpus' forest (~2,700–3,200 m a.s.l.), 'Erica' forest and bushland (~3,200–4,000 m a.s.l.), and alpine 'Helichrysum' shrub vegetation (~3,850–4,600 m a.s.l.). Five study sites per habitat type were distributed to reflect a within-habitat type elevation gradient to detect fine scale changes in biodiversity with changing elevation. Spatial distances among study sites were in all cases larger than 300 m. If possible, study sites were established in core zones of larger areas of the respective habitat type, so that effects of transition zones were minimized.

**Species richness data.** For each taxonomic group, standardized approaches were used to collect species richness data on all 30 study sites. For taxonomic groups where the sampling effort per site was the same, we calculated species richness as the total (cumulative) number of species per study site. These included all groups of vascular plants, terrestrial gastropods, millipedes, spiders, true bugs, Collembola, parasitoid wasps, ants, bees, other aculeate Hymenoptera, ground-dwelling beetles (without dung beetles), dung beetles and birds. Where sampling effort per site varied we used the following approaches: For moths and bats, species richness per site was calculated by averaging the species richness values of the single surveys. For frogs, Zancolli et al.[59] aimed at sampling complete assemblages and therefore sampling effort per site was adjusted to measure asymptotic species richness. The same was true for Orthoptera[60,61]. In both taxa, we used the total number of species per site detected by the authors. In hoverflies numbers of samples varied among study sites. Initially, we reduced the number of samples on sites to a common number and calculated the cumulative species richness from the reduced data set. However, as many species were missing in this data set and the number of species per site was generally low, we run the analysis with the full (unequally sampled) data set and compared the patterns to those of the reduced data set: The pattern was very similar but the variation was much smaller. Therefore, for hoverflies we used the full data set. Details on the sampling procedures for all taxa are described in the Supplementary Methods.

**Temperature and precipitation.** All study sites were equipped with temperature sensors that were installed ~2 m above the ground. Coated plastic funnels were used for radiation shielding. The temperature sensors measured temperatures in 5 min intervals for a time period of ~2 years. We subsequently calculated the MAT as the average of all measurements per study site and the MMT as the average over all monthly temperature minima per study site. On six study sites data loggers were repeatedly stolen and we had to estimate temperature data for these study sites. On the basis of the observed MAT and MMT data we calculated a linear model with MAT/MMT as the response and elevation and habitat type as additive explaining variables ($R^2 = 0.99$, $N = 24$, $P < 0.01$ for both temperature variables). Using this model we predicted the missing six temperature values based on the elevation and habitat type of study sites. MAP was interpolated for every study site using a co-kriging approach based on a 15-year data set from a network of about 70 rain gauges on Mt. Kilimanjaro[55].

**Net primary productivity.** We used the normalized difference vegetation index (NDVI) as a proxy for NPP. Because of the negative biases due to sensor degradation of MODIS Terra[62], NDVI estimations were exclusively based on MODIS Aqua product MYD13Q1 with a horizontal resolution of 250 m × 250 m. Cloud contamination is a very prominent feature on Mt. Kilimanjaro and problematic with regard to realistic estimations of NDVI. To address this issue we first identified all pixels with a MYD13Q1 quality flag of three and deleted these together with the eight adjacent surrounding pixels. Afterwards, we followed the approach proposed by Atzberger and Eilers[63] using the 'Whittaker smoother' based on three iterations with a lambda of 6,000. We then calculated the overall mean NDVI for the 10-year period 2003–2012 and extracted the pixel values corresponding to the locations of our study sites. A comparison of NDVI data with on-site measurements of the leaf-area index conducted during the growing season revealed a high level of correlation ($r = 0.84$, $P < 0.001$).

**Area.** We calculated for each study site the available land area within a range of 300 m above and below the elevational level of study sites. This area data was derived from a digital elevation model of Mt. Kilimanjaro with a resolution of 30 m of the following extent: 37.00074°–37.75602°E, 3.507533°–2.750183°S.

**Mid-domain effect.** We determined for each species the elevational range, which is defined by its minimal and maximum elevation of occurrence. Incomplete sampling routinely underestimates range sizes[64], with the most extreme case being species recorded from only a single elevation (which was the case for many species found) which thus would have an observed elevational range of 0 m. To adjust range underestimation we followed the approach used by Brehm et al.[16] and added 265 m to each end of each recorded range (for all ranges), which corresponds to half the maximum elevational distance between any two adjacent sampling elevations. We then systematically reassigned the location of each of the interpolated, augmented ranges within the domain (606–4,815 m) at random (sampling without replacement) and then recorded the predicted richness for the elevation of each study site. This procedure was repeated for 200 times after which the mean predicted richness for each study site was calculated by averaging over the 200 predicted richness values of the individual repeats.

**Statistical analysis.** Generalized additive models (gam) were used to model the relationships between species richness and elevation[65,66], setting the data family to Gaussian type and the basis dimension of the smoothing function to five. We calculated a complexity value to characterize the shape of elevational species richness patterns with one numeric measure by comparing the explained deviance of generalized linear models (EDglm) with the explained deviance of generalized

additive models (EDgam) of species richness on elevation: complexity = (EDgam − EDglm)/EDgam. Supplementary Fig. 8 exemplifies calculations of complexity for a pattern with a mid-elevational peak, a linear decline and an exponential decline. Note that in case of high error variation, EDglm may theoretically equal EDgam so that complexity values ~0 even though no statistically well supported trend of linear decline is evident. However, in our data set explained deviance was consistently high (>0.3) and all complexity values of <0.2 were associated with significant negative trends (slopes of glm <0, $P < 0.05$). To analyse the statistical support for explanatory variables in predicting plant and animal species richness we used multi-model inference based on information-theory and ordinary least-square regression (20). Multi-model inference not only accounted for uncertainty in parameter estimates but also for uncertainty in model selection[29,30,67]. Moreover, multi-model inference is a way to objectively deal with correlated predictor variables as strongly correlated predictor variables with the same explanatory power will get reduced support in multi-model inference (Supplementary Fig. 9). In the multi-model inference procedure, two final variables were calculated: first, variable importance, which is a measure of the relative support a predictor variable receives over the full model space; second, conditional model-averaged parameter estimates and significance levels. We z-transformed all predictor variables before analyses, so that the model-averaged parameter estimates were standardized beta values. This transformation allowed us to compare the relative influence on species richness among the set of predictor variables which were measured at different scales. For multi-model averaging analyses the R package MuMIn (http://CRAN.R-project.org/package=MuMIn, accessed 25 April 2014) was used.

To analyse elevational species richness patterns and the support for different predictor variables with increasing taxonomic coverage we conducted the following steps: (1) we calculated for all combinations of 1–8 plant ($N = 256$) and for all combinations of 1–16 animal groups ($N = 65,535$; that is, for taxa shown individually in Fig. 1 and Supplementary Fig. 3) the pooled (cumulative) species richness; (2) we modelled pooled species richness as a function of elevation using generalized-additive models and calculated the complexity measure of the elevational species richness pattern; (3) we calculated MDE predictions for all study sites based on the species range information of the randomly combined taxa; and (4) we ran a multi-model inference analysis with pooled species richness as the response and MAT (in animals) or MMT (in plants), MAP, NPP, land area, the predictions of MDE model and PSR (for explaining richness of animal taxa) as predictor variables. These analyses were conducted in the same way as the single taxa analyses.

We validated the influence of incomplete sampling on diversity patterns and inference on predictor variables using a battery of analyses: first, we calculated measures of sample coverage for all taxa using the R package iNEXT[68] and analysed their correlation with elevation. To analyse the influence of incomplete sampling on patterns of elevational diversity, we estimated asymptotic species richness for all single taxa for which abundance data was available, using a non-parametric species richness estimator for abundance-based data, that is, the Chao1 index[27,28,68]. We used generalized additive models to model Chao1-estimated species richness as a function of elevation and predicted for each study site the number of species from these models. We correlated the predicted estimates of asymptotic species richness to the ones predicted by the original model (based on the observed species richness data) using Pearson's product moment correlation. If variation in incomplete sampling significantly biased elevational species richness patterns, we expected predictions not to be highly correlated.

Moreover, as we counted the species of all animal taxa equal in the pooled community-level analyses, results of statistical inference could be biased due to differences in the sampling intensity (that is, here the number of individuals which were sampled per taxon) among taxa. We therefore analysed patterns of elevational species richness and the support for predictor variables additionally for a standardized data set, in which for each animal taxon the number of individuals was rarefied to the lowest number of individuals observed in any of the taxa (that is, $N = 83$) and used for the calculation of variable importance and effect strength of all predictor variables. This procedure was repeated for 5,000 times. From the results of these analyses of standardized data sets we calculated the mean and 95% confidence intervals of variable importance and standardized beta for all predictor variables.

Path analysis[69] was used to disentangle the direct and indirect effect of climate (temperature and precipitation), primary productivity, area, the MDE and plant richness (the latter only for animal species richness as the response variable) on pooled plant and animal species richness. All explanatory variables were standardized by z-transformation using the 'scale' function in R. On the basis of prior studies, we hypothesized that temperature (MAT or MMT) and precipitation (MAP) predict species richness directly and indirectly via their combined effect on primary production (NPP) and on the species richness of plants (PSR, that is, for animal richness as the response variable). Therefore, we pre-selected possible path combinations, by analysing the response variables of our path models (animal species richness, PSR, primary productivity) with all explaining variables:

For plants:

$$NPP \sim MMT + MAP \tag{1}$$

$$PSR \sim MMT + MAP + NPP + Area + MDE \tag{2}$$

For animals:

$$NPP \sim MAT + MAP \quad (3)$$

$$PSR \sim MAT + MAP + NPP + Area + MDE \quad (4)$$

$$Animal\ species\ richness \sim MAT + MAP + NPP + Area + MDE + PSR \quad (5)$$

For each model we used the 'dredge' function of the R package 'MuMIn' to evaluate models defined by all possible variable combinations and ranked them by their AIC-based model weight. As our sample size was relatively low compared with the number of estimated parameters, we used the AIC with a second-order bias correction (AIC$_C$) for inferring the support of individual models. From the set of best supported models ($\Delta$ AIC$_C$ < 3) we calculated paths models using the R package 'lavaan'[70] and ranked them based on their AIC$_C$. In Fig. 5 we show paths coefficients (width of arrows), their statistical significance and multiple coefficients of determination ($R^2$) of predictor variables of the best supported path models with the lowest AIC$_C$. In addition we indicate paths of competitive models ($\Delta$ AIC$_C$ < 3) which were not included in the best model.

**Data availability.** The data that support the findings of this study are available from the following authors upon request: corresponding author (M.K.P.): data of geographic coordinates, elevation, land area, mid-domain effect predictions for study sites, data on Heteroptera, ants, parasitoid and other aculeate wasps; A.H. (andreas.hemp@uni-bayreuth.de): plant data, mean annual precipitation; T.A. (tim.appelhans@staff.uni-marburg.de): NPP, temperature data; A.C. (alice.classen@uni-wuerzburg.de): data on bees and hoverflies; S.W.F. (stefan.ferger@yahoo.de): bird data; S.B.F. (sara.frederiksen@snm.ku.dk): millipede data; F.G. (friederike.gebert@uni-wuerzburg.de): dung beetle data; M.H.-B. (m.helbig-bonitz@gmx.de): bat and moth data; C.H. (claudia.hemp@uni-wuerzburg.de): Orthoptera data; C.N. (cngereza@yahoo.com): gastropod data; J.R. (juliane.roeder@biologie.uni-marburg.de): data on spiders, Collembola, ground-dwelling beetles; G.Z. (giulia.zancolli@gmail.com): amphibian data.

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

## Acknowledgements

We thank the Tanzanian Commission for Science and Technology, the Tanzania Wildlife Research Institute and the Kilimanjaro National Park authority for their support and for granting us access to the Kilimanjaro National Park area. We are grateful to all the companies and private farmers who allowed us to work on their land. We thank the KiLi field staff for helping to collect data at Mt. Kilimanjaro. This study was conducted within the framework of the Research Unit FOR1246 (Kilimanjaro ecosystems under global change: linking biodiversity, biotic interactions and biogeochemical ecosystem processes, https://www.kilimanjaro.biozentrum.uni-wuerzburg.de) funded by the Deutsche Forschungsgemeinschaft (DFG).

## Author contributions

I.S.-D., A.H., M.F. designed the concept for ecological research at Mt. Kilimanjaro. A.H., M.-O.R., M.S., K.B.-G., R.B., E.K.V.K., M.K., T.N., M.T., M.F. and I.S.-D. conceptualized and supervised single taxa studies. A.H. implemented study sites. M.K.P. and I.S.-D. conceived the study. M.K.P., A.H., T.A., C.B., A.C., F.D., A.E., S.B.F., S.W.F., F.G., M.H., M.H.-B., C.H., W.J.K., E.M., C.N., I.O., J.R., G.R., D.S.C., J.T. and G.Z. collected the data. J.D., C.D.E., R.S.P. and A.S. identified significant quantities of specimens. A.H., C.H., V.K. and J.Z. organized and maintained logistic infrastructure. M.K.P. processed and analysed the data and wrote the first version of the manuscript with input from I.S.-D. All authors contributed to the final version of the manuscript.

## Additional information

**Competing financial interests:** The authors declare no competing financial interests.

