## [Peer Review File · Nature Communications]

PEER REVIEW FILE

Reviewers' comments:

Reviewer #1 (Remarks to the Author):

In this study, the authors aim to discover the underlying causes of species richness over an elevational gradient on Mt. Kilimanjaro, for a broad scope of taxa sampling by specialists using a wide variety of taxon-specific techniques. Their key finding is that the environmental correlates of raw species richness differ substantially among individual taxa, but show a single pattern for raw species richness when all taxa are pooled. They report that, for the pooled data, temperature is highly explanatory, which they take as support the Allen et al. (2002, 2006) kinetic energy hypothesis. The only other explanation they consider for elevational patterns of species richness is the "potential energy hypothesis," based on NPP. They found no support for the latter.

I have a number of problems with this study.

(1) It is not the first study to compare the elevational distribution of a wide scope of taxa over a single elevational gradient, but no others are cited or discussed (e.g Colwell et al. 2008, but there are others).

(2) Key references on elevational patterns of richness are neither cited nor discussed, despite their critical relevance to this study. For example, Rahbek (1995, 2005) showed, in a broad meta-analysis, that richness patterns for well-censused, complete gradients were overwhelmingly hump-shaped. How does that square with the claims of generality of temperature (which universally declines with elevation) as driver of diversity in this study? Because the Kilimanjaro gradient studied was not complete (because of habitat loss in the lowlands), the obvious question arises, if the missing lowland biota were still intact, would it be less rich than the lowest point on the sample part of the gradient, forming a hump? If so, then temperature cannot be "driving" richness universally, even on Kilimanjaro.

(3) The authors failed to consider and discuss non-energy-based explanations for elevational patterns of species richness. The most obvious omission is elevational band area, per se, not band

area multiplied by NPP (which was considered, but is strongly shaped by NPP and confounds area-per-se with NPP). This regional or "indirect" area effect on local richness (Romdal and Grytnes 2007) cannot be dismissed out-of-hand, and is routinely included in multivariate studies of the correlates of species richness on elevational gradients. Last time I saw Mt. Kilimanjaro it was rather conical, which means that temperature is closely correlated with elevational band area. Statistics are required, of course, but my guess is that there will be no way to exclude the indirect area effect as an explanation for the richness pattern over the sampled elevations. That would profoundly change the conclusions drawn in this study. The other principal non-energy-based factor is the mid-domain effect, but it cannot be evaluated for incomplete gradients.

(4) It is statistically inevitable that pooling discordant datasets will produce a pattern for the pooled data that differs from some or all of those patterns that best describe the underlying individual datasets.

(5) Because the authors counted all species as equal in the pooled data, the elevational richness pattern shown by pooled data is a weighted average of the patterns shown by the taxon-specific datasets. Thus, not only underlying differences in species richness, but differences in inventory completeness (statistical coverage...more on that later) and the degree to which inventoried groups are representative will substantially affect the outcome. Of the 9 groups with the largest number of recorded species in this study, 8 show a declining pattern of richness with elevation ("beetles" are the exception) and together account for 70% of the pooled species. Is it any wonder that the pooled pattern has a declining pattern of richness with elevation?

To my mind, it would make more sense to average the spatial patterns of richness of the different groups in a way that each group (not each species) contributes equally to the pooled pattern. The most rigorous way to do this would be by rarefaction (resampling). If the smallest taxonomic group has n individuals, then pool n individuals at random from each group, repeat many times, and analyze the mean pattern of richness among the resamples, with confidence intervals based on variance among the resamples.

(6) As the authors note, niche conservatism tends to limit the elevational distribution of related species (though the relevant literature for phylogenetic conservatism for elevation is not cited, e.g. Wu et al. 2013, among others). This means that elevational locations are phylogenetically non-independent, within taxa: all the more reason to average patterns among taxa, not among individuals.

(7) Using raw species counts from sampling tropical biotas in small plots over limited periods of time is a certain recipe for undersampling bias, especially for hyper-diverse groups like insects. The notions of "sampling intensity" and "completeness" are never defined in this study, nor in any of the three papers cited on Line 241 (unless eyeballing a sequential accumulation curve, as

in the Hemp papers, is a measure of "completeness).

Let's define "inventory completeness" by its widely accepted current definition, sample coverage (Chao et al. 2014). Neither equal-sized plots nor equal time searching or equal numbers of samples guarantees equal inventory completeness. Individuals, not plots or time units, carry the information of species identity. Even if equal-sized plots (or equal time searching or equal numbers of samples) produced the same average number of individuals at each elevation on an elevational transect (which is very often not the case!), poorer assemblages are better censused than richer ones, for the same number of individuals, unless an asymptote has been reached.

Aware of the problem of undersampling bias, the authors attempt to reassure us by showing that the results using Chao1 richness estimator are well correlated with results using raw richness values. Chao1 is rigorous richness estimator, but it estimates minimum richness, given the data, so that the estimated richness is often much less than true richness for diverse and/or undersampled taxa. (The authors cite not a single paper on richness estimation, not even the 30+ year old paper that introduced Chao1.) Instead, inventory completeness should be documented as sample coverage, and coverage-based rarefaction and extrapolation could be used to compare richness among all groups (Chao et al. 2014).

(8) The MS rightly touts the wide taxonomic scope of this study, but I think the pie chart in Fig. 1 may be misleading. The caption reads, "pie charts show the approximate contribution of the studied higher taxa to the described." Does that mean, for example, that the few groups of beetles that can be collected in pitfalls stand in for the immense diversity of tropical Coleoptera, and is it further claimed that their elevational distribution on this one mountain fairly represents all beetles?

References cited in this review but not in the MS:

Chao, A., Gotelli, N.J., Hsieh, T.C., Sander, E.L., Ma, K.H., Colwell, R.K. et al. (2014).

Rarefaction and extrapolation with Hill numbers: a framework for sampling and estimation in species diversity studies. *Ecol. Monogr.*, 84, 45-67.

Colwell, R.K., Brehm, G., Cardelús, C., Gilman, A.C. & Longino, J.T. (2008). Global warming, elevational range shifts, and lowland biotic attrition in the wet tropics. *Science*, 322, 258-261.

Rahbek, C. (1995). The elevational gradient of species richness: a uniform pattern? *Ecography*, 19, 200-205.

Rahbek, C. (2005). The role of spatial scale in the perception of large-scale species-richness patterns. *Ecol. Lett.*, 8, 224-239.

Romdal, T.S. & Grytnes, J.A. (2007). The indirect area effect on elevational species richness patterns. *Ecography*, 30, 440-448.

Wu, Y., Colwell, R.K., Han, N., Zhang, R., Wang, W., Quan, Q. et al. (2014). Understanding historical and current patterns of species richness of babblers along a 5000-m subtropical elevational gradient. *Global Ecol. Biogeogr.*, 1167-1176

Robert K. Colwell
University of Connecticut

Reviewer #2 (Remarks to the Author):

Determinants of elevational biodiversity gradients change from single taxa to multi-taxa...

Peters et al. for *Nature Communications*

What a tour de force of a field campaign. As Peters and colleagues point out, there's almost nothing quite like this study in the literature. I'll summarize a huge amount of work in just a couple of sentences: Peters and colleagues sampled 30 sites along the extensive elevational gradient on Mt Kilimanjaro. At each of those sites, they quantified diversity of 21 different plant and animal taxa. The key result is that the 21 different taxa show a variety of elevational gradients, but when you pool them all together, they show a pretty striking (mostly) linear decline in diversity with increasing elevation.

This is a great paper, and certainly will become an instant classic in biogeography, macroecology, and biodiversity studies. But I did have some thoughts on how it could be made (perhaps) better:

Thought 1: For the most part, the taxa for which biodiversity declines linearly are those with a lot of species on Mt Kilimanjaro. For example - the grasshoppers and birds and what I think are the roses and monocots, maybe? have incredibly high richness at the lowest elevations. Could it be that the low diversity taxa are hump-shaped because they are endemics? Or don't have species whose ranges extend out into the lowland sites surrounding Kilimanjaro? Is there any way to test that? For instance - look at the bees. There are those 5 low-elevation sites where diversity is really really high, then diversity drops off. Is diversity inflated at those low elevations because most low-elevation species have populations that extend out into the lowlands? And, because there are these few examples of the most diverse taxa exhibiting these linear declines, when you

put the diverse taxa together with the low-diversity taxa, the effect of the high diversity taxa swamps any pattern of the low diversity taxa.

Thought 2: And continuing to look at the bee figure, it looks like those 5 or so low-elevation sites are really high in diversity, but then there's no relationship between diversity and elevation at the highest 25 sites. In fact, that kind of pattern emerges for several taxa (the hump-shaped pattern of the flies, I guess, the orthopterans, the collembolans, parasitoid wasps, amphibians, monocots, and ferns. So, is every pattern driven just by what's happening at those 5 low-elevation sites, that is either really low for some taxa (e.g., ferns), or really high (e.g., bees)? Could explaining why diversity varies between those sites and the rest of the gradient for many taxa be the explanation for the macroecological pattern?

Thought 3: While I am a huge fan of working across spatial and taxonomic scales, I don't necessarily agree with these two key sentences: "Our findings show the value of multi-taxa studies to identify general models of diversity gradients and underscore the importance of temperature-dependent evolutionary and ecological processes for diversification and species coexistence," and "our study revealed that a broad taxonomic coverage in macroecological studies provides new insights into the drivers of broad-scale diversity gradients." First of all, how does it show the value? Because the explanatory value goes up if you include more different kinds of species? And what new insights? That diversity increases with temperature? And I also don't agree that this work says much about temperature-dependent evolutionary processes like diversification. How does it? Which of these lineages has shown higher diversification at lower elevations on Mt Kilimanjaro? Which has shown lower diversification at lower temperatures? What is the relationship between diversification and temperature for any of these taxa? R

And this kinetic-based theory, has moved on to basically say that temperature and productivity are correlated with diversity. It's no longer just about how temperature and temperature alone drives mutation rate and in turn diversification. And I think this is where this paper falls short - just showing that diversity is correlated with temperature is not the same thing as supporting a kinetic-based theory of diversity. I think stronger support would come from actually honing in on some mechanisms or links between temperature and diversity, not just saying that temperature is correlated with diversity, ergo, kinetics. See for example Sibley et al.'s edited volume for updated versions.

Thought 4: There are a number of papers that have examined 'taxonomic scaling' in a variety of ways, such as by asking whether the number of species across communities is correlated with the number of genera or families across those same communities (see Enquist et al. 2007 *International Journal of Plant Science* 168: 729-749 for an example). To my recollection, most of those studies show that the number of species, genera, families, etc. are often very strongly correlated (e.g., correlation coefficients are often >0.9). But here, in some ways, you're not

finding that, because as you move from species to everything, the pattern is reversed. Why not include the intermediate taxonomic levels as well? That is, plot number of species, number of genera, number of families, number of orders, etc. against elevation and temperature.

Thought 5: Why not do some sort of structural equation modeling?

Response to referee comments

General response: Many thanks for conducting the constructive reviews on our manuscript and providing many helpful comments to improve it. We addressed all comments, set up new analyses and rewrote the manuscript significantly. Additionally, we integrated data for four more groups of animals (true bugs, millipedes, spiders and dung beetles) for which identification has been finished while the manuscript was in review. Integrating these additional groups supports the major results and conclusions of the study and makes the reported patterns even more clear.

Dear Dr Peters,

Your manuscript entitled "Determinants of elevational biodiversity gradients change from single taxa to the multi-taxa community level" has now been seen by 2 referees, whose comments are appended below. I sincerely apologize for the unusual delay in the review process. In addition the initial difficulty finding available referees, I'm afraid that the editorial team has undergone a transition recently and we are short-staffed as consequence.

While the referees find the underlying dataset and the work of potential interest, they have raised substantive concerns that in our view need to be addressed before we can consider publication in Nature Communications. As you will see, both referees raise somewhat overlapping concerns relating to the analyses, scholarship and presentation of the study. As such, should further analysis and textual revision allow you to address these criticisms, we would be happy to look at a revised manuscript.

However, I should stress that, because Nature Communications strives to provide an efficient editorial service and fast publication, we are reluctant to see manuscripts undergo multiple rounds of review. As such, we would recommend that extensive revision is made to address all of these concerns in full before resubmission. If the revision process takes significantly longer than three months, we will be happy to reconsider your paper at a later date, as long as nothing similar has been accepted for publication at Nature Communications or published elsewhere in the meantime.

When resubmitting your paper, we also ask that you ensure that your manuscript complies with our editorial policies.

Specifically, please ensure that the following requirements are met, and any relevant checklists are completed and uploaded with the revised article:

In the meantime, we hope you will find our referees' comments helpful. Please do not hesitate to contact us if there is anything you would like to discuss. Again, I am very sorry about the delay in the process and please extend my apologies to your co-authors as well.

Best regards,
Min

Min Cho, PhD
Senior Editor
Nature Communications
orcid.org/0000-0003-4696-0173
<http://www.nature.com/ncomms/index.html>

Reviewers' comments:

Reviewer #1 (Remarks to the Author):

In this study, the authors aim to discover the underlying causes of species richness over an elevational gradient on Mt. Kilimanjaro, for a broad scope of taxa sampling by specialists using a wide variety of taxon-specific techniques. Their key finding is that the environmental correlates of raw species richness differ substantially among individual taxa, but show a single pattern for raw species richness when all taxa are pooled. They report that, for the pooled data, temperature is highly explanatory, which they take as support the Allen et al. (2002, 2006) kinetic energy hypothesis. The only other explanation they consider for elevational patterns of species richness is the "potential energy hypothesis," based on NPP. They found no support for the latter. I have a number of problems with this study.

(1) It is not the first study to compare the elevational distribution of a wide scope of taxa over a single elevational gradient, but no others are cited or discussed (e.g. Colwell et al. 2008, but there are others).

Response: Prof. Colwell points out that there are other papers reporting patterns of elevational diversity for more than a single taxon, including his classical paper in *Science* from 2008 (surely we know this!). However, these papers are either not focused on patterns of elevational diversity (e.g. the 2008 paper in *Science* is on impact of climatic warming on elevational range shift) or they were based on a much smaller set of taxa (this is the case for all former analyses). We critically surveyed again the literature and added sentences in the discussion in which we name former multi-taxa studies and explain why we are still convinced that our manuscript is novel regarding the range of taxa included and the innovative way how we analyze diversity patterns from single taxa to multi-taxa communities. We think that the standardized, well replicated assessment of the large number of taxa, spanning the whole animal and vascular plant tree of life, contrasts strongly to former analyses which were either focused on specific smaller clades (e.g. vertebrates) or, when including multiple clades, included only a small number of taxa (typically not more than 3 or 4). The number of taxa included in our analyses is high enough to make generalizations which could not been done with more restricted data sets used in former studies.

(2) Key references on elevational patterns of richness are neither cited nor discussed, despite their critical relevance to this study. For example, Rahbek (1995, 2005) showed, in a broad meta-analysis, that richness patterns for well-censused, complete gradients were overwhelmingly hump-shaped. How does that square with the claims of generality of temperature (which universally declines with elevation) as driver of diversity in this study? Because the Kilimanjaro gradient studied was not complete (because of habitat loss in the lowlands), the obvious question arises, if the missing lowland biota were still intact, would it be less rich than the lowest point on the sample part of the gradient, forming a hump? If so, then temperature cannot be "driving" richness universally, even on Kilimanjaro.

Response: (a) In the discussion of our manuscript we now incorporated more key references and discussed them with the findings of our study. Rahbek (1995, 2005) indeed showed that a majority of distributions are hump-shaped. However, please note that we also found a high proportion of hump-shaped patterns at the level of single taxa (N = 4 of 8 plant taxa, 5 of 12 animal taxa; with the new data 7 of 16 animal taxa) but that at higher levels of taxonomic coverage patterns exclusively tend to be of monotonous decline because more taxa and

particularly those contributing larger numbers of species are found in habitats of high temperature. The metaanalyses of Rahbek (1995, 2005) did not evaluate how the taxonomic scale or coverage influenced patterns of elevational diversity. Under the assumption that environmental niches of species are inherited and change slowly over evolutionary time, smaller clades (at the extreme end: a single species) will tend to show higher proportions of unimodal patterns of diversity than larger clades for which phylogenetic autocorrelation in niches is less pronounced.

(b) Concerning the comment that the gradient was not complete due to habitat loss in the lowlands: This is a misunderstanding we clarified in the revised version of the manuscript. The lower limit of the elevational gradient was not constrained by missing natural habitat at the base of the mountain but by missing elevations between 0 and 800 m. In fact our natural habitat at the lowest elevations can be considered as rather typical for East Africa where most of the land area is situated at elevations >700 m asl. This includes e.g. the famous Serengeti, Masai Mara, Amboseli and the largest savannah park in Kenya, the Tsavo NP (the latter is directly adjacent to some of our savannah sites). Within a square of 2° x 2° (ca. 222 x 222 km) with the Kibo peak of Mt. Kilimanjaro in the center, 100% of the area is >500 m asl, 99% >600 m asl, 94% >700 m asl and 83% is >800 m asl. We added sentences in the method section to make this clear.

(3) The authors failed to consider and discuss non-energy-based explanations for elevational patterns of species richness. The most obvious omission is elevational band area, per se, not band area multiplied by NPP (which was considered, but is strongly shaped by NPP and confounds area-per-se with NPP). This regional or "indirect" area effect on local richness (Romdal and Grytnes 2007) cannot be dismissed out-of-hand, and is routinely included in multivariate studies of the correlates of species richness on elevational gradients. Last time I saw Mt. Kilimanjaro it was rather conical, which means that temperature is closely correlated with elevational band area. Statistics are required, of course, but my guess is that there will be no way to exclude the indirect area effect as an explanation for the richness pattern over the sampled elevations. That would profoundly change the conclusions drawn in this study. The other principal non-energy-based factor is the mid-domain effect, but it cannot be evaluated for incomplete gradients.

Response: Prof. Colwell criticizes that we missed to include important non-energy explanations for elevational patterns of species richness. He mentions particularly two potential predictors of species richness, area and the mid-domain effect. First, we have to state that there was a misunderstanding possibly caused by our first introductory paragraph which only contrasts two energy-based hypotheses. In our original analyses we included three energy based hypotheses (mean annual temperature, local NPP, elevational belt NPP) but also one (plants) or two (animals) non-energy based hypotheses (mean annual precipitation and for animals, additionally, plant species richness). We rewrote the introduction to make this clear. Second, in the new version of our manuscript we included more non-energy based hypotheses in the models, i.e. the area and the mid-domain effect hypotheses. We included the mid-domain effect because, unlike assumed by Prof. Colwell (see last comment), the gradient was not incomplete. The lowest elevation which we used can be considered as a typical base elevation of the mountain and the mid-domain effect has been successfully tested repeatedly with similar kind of elevational gradients (e.g. McCain 2005 Ecology, Bauer et al. 2014 Ecography). Concerning area as a predictor variable: while temperature declines quasi linearly with elevation area shows a negative exponential relationship. Moreover, as most parts of East Africa, including the Kilimanjaro region, forms a plateau of >700 m elevation, the increase of area with decreasing elevation levels off between 700 and 1100 m asl.

Therefore, we were able to statistically separate potential effects of temperature and area on elevational diversity patterns. Due to a very strong correlation of elevational belt NPP and area (pearson's $r = 0.98$) we only used area and local NPP as explanatory variables and deleted elevational belt NPP from models. The major results of the manuscript remain unaffected by including area or mid-domain effect predictions as explanatory variables.

(4) It is statistically inevitable that pooling discordant datasets will produce a pattern for the pooled data that differs from some or all of those patterns that best describe the underlying individual datasets.

Response: Even though we fully agree that a combination of discordant datasets may produce patterns differing from single data sets, our analyses show that on the way to the full taxonomic coverage of producer and consumer assemblages there is a strong tendency for temperature becoming a single and strong predictor of diversity. This result is independent from the identity of the taxa incorporated in analyses: Taking out any five animal taxa from the analyses will reveal mean annual temperature as the most important predictor of diversity in terms of variable importance and strength of effect in 98% of all taxa combinations. To make this clear we changed Figure 4 of the manuscript in a way that not only the means but the full variation in variable importance and standardized beta values are shown for each variable. Importantly, the patterns seen at the highest level of taxonomic coverage represent not an arbitrary species richness pattern constructed by artificially combining some taxa but show trends at the level of local plant and animal communities: The plant censuses were complete and the species richness in each study site were accurate, such that species richness of local plant communities can be assumed to be well represented. Even though the animal data included large parts of the major terrestrial taxa from the whole animal phylogenetic tree it was not complete. However, our sampling resembles a stratified random sampling procedure (first strata: selection of taxa, second strata: sampling of species within taxa from the local assemblages) which allows comparative (relative) estimates of animal community richness even under incomplete sampling of taxa and species within taxa. We added a new Supplementary Figure 7 on this topic.

(5) Because the authors counted all species as equal in the pooled data, the elevational richness pattern shown by pooled data is a weighted average of the patterns shown by the taxon-specific datasets. Thus, not only underlying differences in species richness, but differences in inventory completeness (statistical coverage...more on that later) and the degree to which inventoried groups are representative will substantially affect the outcome. Of the 9 groups with the largest number of recorded species in this study, 8 show a declining pattern of richness with elevation ("beetles" are the exception) and together account for 70% of the pooled species. Is it any wonder that the pooled pattern has a declining pattern of richness with elevation?

To my mind, it would make more sense to average the spatial patterns of richness of the different groups in a way that each group (not each species) contributes equally to the pooled pattern. The most rigorous way to do this would be by rarefaction (resampling). If the smallest taxonomic group has n individuals, then pool n individuals at random from each group, repeat many times, and analyze the mean pattern of richness among the resamples, with confidence intervals based on variance among the resamples.

Response: Thanks for this suggestion. This was clearly missing! We followed the advice of Prof. Colwell and used a rarefaction approach to analyze whether differences in sampling among taxonomic groups biased results. This strengthens the results by removing a potential

bias caused by inhomogeneous sampling of taxa. We repeatedly ($N = 2000$) sampled randomly 83 individuals (the lowest number of pooled individuals observed for any of the taxa shown in Figure 1) from each taxonomic group and calculated with this rarefied data set patterns of elevational diversity and the support for explanatory variables. This approach led to the same conclusions as the analyses with the full data set. We present the results in a new supplementary file (Supplementary Fig. 5).

(6) As the authors note, niche conservatism tends to limit the elevational distribution of related species (though the relevant literature for phylogenetic conservatism for elevation is not cited, e.g. Wu et al. 2013, among others). This means that elevational locations are phylogenetically non-independent, within taxa: all the more reason to average patterns among taxa, not among individuals.

Response: Thanks for this comment. We incorporate additional literature, including the study cited. We ran all analyses with standardized data sets (see response to the last comment) and the results remained consistent.

(7) Using raw species counts from sampling tropical biotas in small plots over limited periods of time is a certain recipe for undersampling bias, especially for hyper-diverse groups like insects. The notions of "sampling intensity" and "completeness" are never defined in this study, nor in any of the three papers cited on Line 241 (unless eyeballing a sequential accumulation curve, as in the Hemp papers, is a measure of "completeness").

Let's define "inventory completeness" by its widely accepted current definition, sample coverage (Chao et al. 2014). Neither equal-sized plots nor equal time searching or equal numbers of samples guarantees equal inventory completeness. Individuals, not plots or time units, carry the information of species identity. Even if equal-sized plots (or equal time searching or equal numbers of samples) produced the same average number of individuals at each elevation on an elevational transect (which is very often not the case!), poorer assemblages are better censused than richer ones, for the same number of individuals, unless an asymptote has been reached.

Aware of the problem of undersampling bias, the authors attempt to reassure us by showing that the results using Chao1 richness estimator are well correlated with results using raw richness values. Chao1 is rigorous richness estimator, but it estimates minimum richness, given the data, so that the estimated richness is often much less than true richness for diverse and/or undersampled taxa. (The authors cite not a single paper on richness estimation, not even the 30+ year old paper that introduced Chao1.) Instead, inventory completeness should be documented as sample coverage, and coverage-based rarefaction and extrapolation could be used to compare richness among all groups (Chao et al. 2014).

Response: We are aware of the problem of the undersampling bias and, indeed, as Prof. Colwell already points out, we addressed the problem by comparing the raw data of species richness with Chao 1 richness estimators which revealed a very high accordance with results based on the original data. In the new version of the manuscript we strengthened the text pointing to the potential problems of variation in sampling completeness. We cited the old paper in which the Chao1 index is introduced and newer papers on the topic (e.g. Chao et al. 2014), and documented values of sample coverage for all data sets in the new Supplementary Table 1.

(8) The MS rightly touts the wide taxonomic scope of this study, but I think the pie chart in Fig. 1 may be misleading. The caption reads, "pie charts show the approximate contribution of the studied higher taxa to the described." Does that mean, for example, that the few groups of beetles that can be collected in pitfalls stand in for the immense diversity of tropical Coleoptera, and is it further claimed that their elevational distribution on this one mountain fairly represents all beetles?

Response: We see the problem readers might have with this figure. We deleted it in the new version of the figure.

References cited in this review but not in the MS:

Chao, A., Gotelli, N.J., Hsieh, T.C., Sander, E.L., Ma, K.H., Colwell, R.K. et al. (2014). Rarefaction and extrapolation with Hill numbers: a framework for sampling and estimation in species diversity studies. *Ecol. Monogr.*, 84, 45-67.

Colwell, R.K., Brehm, G., Cardelús, C., Gilman, A.C. & Longino, J.T. (2008). Global warming, elevational range shifts, and lowland biotic attrition in the wet tropics. *Science*, 322, 258-261.

Rahbek, C. (1995). The elevational gradient of species richness: a uniform pattern? *Ecography*, 19, 200-205.

Rahbek, C. (2005). The role of spatial scale in the perception of large-scale species-richness patterns. *Ecol. Lett.*, 8, 224-239.

Romdal, T.S. & Grytnes, J.A. (2007). The indirect area effect on elevational species richness patterns. *Ecography*, 30, 440-448.

Wu, Y., Colwell, R.K., Han, N., Zhang, R., Wang, W., Quan, Q. et al. (2014). Understanding historical and current patterns of species richness of babblers along a 5000-m subtropical elevational gradient. *Global Ecol. Biogeogr.*, 1167-1176

Robert K. Colwell
University of Connecticut

Response: Many thanks for the advice on these papers. We incorporated them in the new version of the manuscript.

Reviewer #2 (Remarks to the Author):

Determinants of elevational biodiversity gradients change from single taxa to multi-taxa...

Peters et al. for Nature Communications

What a tour de force of a field campaign. As Peters and colleagues point out, there's almost nothing quite like this study in the literature. I'll summarize a huge amount of work in just a couple of sentences: Peters and colleagues sampled 30 sites along the extensive elevational gradient on Mt Kilimanjaro. At each of those sites, they quantified diversity of 21 different plant and animal taxa. The key result is that the 21 different taxa show a variety of elevational

gradients, but when you pool them all together, they show a pretty striking (mostly) linear decline in diversity with increasing elevation.

This is a great paper, and certainly will become an instant classic in biogeography, macroecology, and biodiversity studies. But I did have some thoughts on how it could be made (perhaps) better:

Response: Many thanks for this comment and the nice words.

Thought 1: For the most part, the taxa for which biodiversity declines linearly are those with a lot of species on Mt Kilimanjaro. For example - the grasshoppers and birds and what I think are the roses and monocots, maybe? have incredibly high richness at the lowest elevations. Could it be that the low diversity taxa are hump-shaped because they are endemics? Or don't have species whose ranges extend out into the lowland sites surrounding Kilimanjaro? Is there any way to test that? For instance - look at the bees. There are those 5 low-elevation sites where diversity is really really high, then diversity drops off. Is diversity inflated at those low elevations because most low-elevation species have populations that extend out into the lowlands? And, because there are these few examples of the most diverse taxa exhibiting these linear declines, when you put the diverse taxa together with the low-diversity taxa, the effect of the high diversity taxa swamps any pattern of the low diversity taxa.

Response: (a) The distribution of most of the species recorded on Mt. Kilimanjaro are unknown which makes it hard to infer relationships between patterns of elevational diversity, distribution of species in the lowlands and endemism. Even for the taxa from Mt. Kilimanjaro where lists about endemism exist (e.g. gastropoda) we suspect them as problematic as several large mountains in Tanzania, Kenya and Uganda remain little studied or were not studied at all. Several taxa had to be identified to family and morphospecies level only (e.g. the most species rich group, i.e. the parasitoid hymenoptera) as taxonomy is poorly developed and many species are probably not even described. When it comes to distributional data the situation is even much worse. The comment of reviewer 2, however, indirectly suggests that patterns of elevational diversity could be determined by available land area with more species occurring in the extensive lowlands than in the area-restricted highlands. In the new version of the manuscript we included area as an explanatory variable. However, land area appeared to be less supported as a predictor of species richness than temperature, for single taxa but also for the full assemblage.

(b) Reviewer 2 criticizes that putting high diversity taxa together with low diversity taxa swamps any pattern of the low diversity taxa. This is correct as an increase in diversity may be produced by a higher number of higher level taxonomic units (orders, families, genera) but also by a higher level of species per taxonomic unit. We regard this not as a problem but think that the diversity of the whole animal assemblage is an important and interesting parameter per se (similarly as bird species richness is of interest and not only the species richness of subclades like thrushes). Please also see that the differences in species richness among taxa are moderate with the maximum species richness of the taxon with the highest site richness (moths) only being 7.5x higher than the richness of the taxon with the lowest site richness (true bugs). In order to control for differences in sampling efforts among taxa we now included new analyses in which we standardized the sampling effort for all taxa (based on the randomized sampling of a common number of individuals per taxon). These analyses revealed the same results as the full data set (see new Supplementary Fig. 5).

Additionally, please note that the results are quite robust against the exclusion of taxa with high diversity: when we take out the two taxa with highest site diversity from the data set

(moths, Orthoptera) elevational diversity is still linearly declining and mean annual temperature is the best predictor of species richness in terms of both effect strength (std beta = 0.78) and variable importance (= 1). When we take out the three most diverse groups (- Orthoptera, -moths, -birds) elevational diversity is still linearly declining (explained deviance = 84%) and temperature has the highest effect strength (std beta = 0.73) and the second highest variable importance (VI = 0.7) (npp has a higher variable importance (=0.9) but lower effect strength (std beta = 0.33). When the four most diverse groups are removed, pattern of elevational diversity is linearly declining (explained deviance = 89.3%) and temperature has the highest effect strength (std beta = 0.84) and variable importance (VI = 1) (std beta and VI of npp = 0.18 and 0.84). So the animal community level patterns are not really depending on the highly diverse taxa. This is shown in a general way in the new Figure 4 where we show the full variation in variable importance and standardized beta values for all predictor variables for a sequential removal of 0 to 15 taxa.

Thought 2: And continuing to look at the bee figure, it looks like those 5 or so low-elevation sites are really high in diversity, but then there's no relationship between diversity and elevation at the highest 25 sites. In fact, that kind of pattern emerges for several taxa (the hump-shaped pattern of the flies, I guess, the orthopterans, the collembolans, parasitoid wasps, amphibians, monocots, and ferns. So, is every pattern driven just by what's happening at those 5 low-elevation sites, that is either really low for some taxa (e.g., ferns), or really high (e.g., bees)? Could explaining why diversity varies between those sites and the rest of the gradient for many taxa be the explanation for the macroecological pattern?

Response: Even though we see that the lower five sites are of large influence for the pattern it is definitely not the case that there is no relationship between diversity and elevation for the remaining 25 sites. Most of the detected patterns are still evident for data sets in which the lowland savannah habitats (N = 5 study sites) were removed from analyses. For example, for bees, a taxon mentioned by reviewer 2, a negative relationship between elevation and species richness still holds even when the five lowest sites are removed (explained deviance = 25.7%, edf=1, n = 25, p = 0.00954). The same is true for the orthopterans (ED = 68.6%, edf = 1.86, n = 25, p < 0.0001), and frogs (ED = 31.7%, edf = 1, n = 14, p = 0.035). Likewise, removing the five lowest sites for taxa with unimodal (hump-shaped) diversity patterns in many cases does not lead to a loss of the unimodal pattern (e.g. in ferns, magnoliids, gastropods). Only in taxa where the peak of the unimodal distribution is located in the lower quarter of the elevational gradient (e.g. in Collembola, parasitoid wasps) the exclusion of savannah sites leads to a monotonous decline of species richness with elevation.

Some taxa have much higher species richness at the lowest five sites than at the next five study sites along the elevational gradient (those with exponential type patterns of elevational diversity). In other taxa, however, the decline was linear. Similarly, in some taxa with a unimodal distribution of diversity species richness declines at the lowest elevations but there are also taxa where the decline already starts in lower montane forests. The peak of unimodal distributions is highly variable as Supplementary Fig. 4 shows. With these words we want to make clear that there is not a single universal threshold elevation where diversity either declines or increases but the threshold is rather idiosyncratic.

Thought 3: While I am a huge fan of working across spatial and taxonomic scales, I don't necessarily agree with these two key sentences: "Our findings show the value of multi-taxa studies to identify general models of diversity gradients and underscore the importance of temperature-dependent evolutionary and ecological processes for diversification and species coexistence," and "our study revealed that a broad taxonomic coverage in macroecological

studies provides new insights into the drivers of broad-scale diversity gradients." First of all, how does it show the value? Because the explanatory value goes up if you include more different kinds of species? And what new insights? That diversity increases with temperature? And I also don't agree that this work says much about temperature-dependent evolutionary processes like diversification. How does it? Which of these lineages has shown higher diversification at lower elevations on Mt Kilimanjaro? Which has shown lower diversification at lower temperatures? What is the relationship between diversification and temperature for any of these taxa? R

And this kinetic-based theory, has moved on to basically say that temperature and productivity are correlated with diversity. It's no longer just about how temperature and temperature alone drives mutation rate and in turn diversification. And I think this is where this paper falls short - just showing that diversity is correlated with temperature is not the same thing as supporting a kinetic-based theory of diversity. I think stronger support would come from actually honing in on some mechanisms or links between temperature and diversity, not just saying that temperature is correlated with diversity, ergo, kinetics. See for example Sibley et al.'s edited volume for updated versions.

Response: Reviewer 2 raised three interesting points focusing on (1) the value of the multi-taxa approach in comparison to a traditional single taxa approach, (2) the lack of direct support for a kinetic-based theory of diversity and (3) additional analyses to test mechanistic links between temperature and diversity. Concerning point (1): How does it show the value: First, the analyses of single taxa along the same environmental gradient reveals support for different hypothesis to explain species richness. This basically gives some support to the view that a general, cross-taxon driver of diversity is unlikely to be found for the clades which are typically studied in macroecology (e.g. birds, vertebrates, ants). We consider this a first valuable result which could be anticipated but has not been shown before with a comparable taxonomic and environmental extend which we used in our study. Second, we show that the support for drivers of diversity are depending on the taxonomic coverage of the study. We consider this also an important result. Scaling up from regional/continental to global scales has shifted the relative support for different hypotheses to explain diversity gradients. Similarly, our study shows that the answer to the question what drives diversity depends on the taxonomic scale of analysis. As we point out in the introduction, hypotheses like the productivity hypothesis or the hypothesis relating the variation in species richness to kinetic energy are assumed to be depending on the taxonomic inclusiveness (see e.g. Hurlbert and Stegen 2014) but this has not been tested with quantitative data before. Third, at the highest level of taxonomic coverage we find high support for temperature as a major driver of species richness. Temperature has been revealed as an important driver of diversity also in other studies but in our manuscript we show its importance at the level of plant and animal communities. We want to point out that it is not trivial to find the support for temperature at this level of taxonomic coverage as single taxa analyses regularly reveal support for different hypothesis to explain diversity gradients. We changed considerable parts of the introduction, results, figures and discussion and hope that the added value of a multi-taxa approach becomes more visible.

Concerning point (2): We fully agree here. We rewrote parts of the introduction and discussion in a way that the finding of a positive temperature-richness relationship is discussed in a more general way. Concerning point (3): Here we also fully agree. Identifying the potential links between temperature and diversity is of high importance and we did this already for single taxa like bees (Classen et al. 2015, *Global Ecology and Biogeography*) or birds (Ferber et al. 2014, *Global Ecology and Biogeography*). However, testing these potential links requires time-intensive collection of additional data (e.g. resource abundance, species

interaction rates, biomass, or DNA data) which is currently only available for few taxa in the data set. Any analyses with this additional data would have to be restricted to a minor subset of the taxa included in the manuscript and we think that this would not fit well to the overall scope of the manuscript. We think that the strength of the study is the large number of taxa.

Thought 4: There are a number of papers that have examined 'taxonomic scaling' in a variety of ways, such as by asking whether the number of species across communities is correlated with the number of genera or families across those same communities (see Enquist et al. 2007 International Journal of Plant Science 168: 729-749 for an example). To my recollection, most of those studies show that the number of species, genera, families, etc. are often very strongly correlated (e.g., correlation coefficients are often >0.9). But here, in some ways, you're not finding that, because as you move from species to everything, the pattern is reversed. Why not include the intermediate taxonomic levels as well? That is, plot number of species, number of genera, number of families, number of orders, etc. against elevation and temperature.

Response: We added figures showing trends in species richness, family richness and order richness for plants and animals along the elevational gradient (Figure 3). They show principally similar trends like species richness but that the increase of species richness in the lowlands is mainly driven by more species per family/order rather than by increasing numbers of higher level taxa. However, even though we think these trends are interesting, one has to be careful with any further interpretation because (a) for several higher level clades some subclades are missing and, (b), most important the taxonomic levels are subjective concepts with often little comparability, at least in animals. For example, the birds in our data belong to a clade not older than 130 mio years which are composed of 45 families. In contrast, the ants have a similar age but all species belong to a single family. Due to these problems, in the new version of the manuscript we simply showed trends for higher level taxa but we did not conduct any further statistical analyses like relating taxon richness to any of the explanatory variables. With good backbone of molecular data at hand this would provide a very interesting analysis.

Thought 5: Why not do some sort of structural equation modeling?

Response: Thanks for this advice. In the revised version of the manuscript we used the lavaan package in R to calculate path models to represent the direct and indirect effects of predictor variables on animal and plant diversity (new Fig. 5). Several of the used predictor variables are indeed interrelated (in particular temperature and precipitation as drivers of primary productivity) and these relationships are now well reflected in the paper.

Reviewers' comments:

Reviewer #1 (Remarks to the Author):

I commend the authors for undertaking extensive additional analyses, some quite original, in response to reviewers' comments, including my own, and for moderating over-interpretation of the results as support for the "kinetic energy hypothesis." The study is now better integrated with the existing literature. Overall, it is much strengthened, and certainly deserves to be published.

But I still have concerns, especially about the interpretation and presentation of the main result: that temperature actually "drives" overall species richness of multiple clades (or slightly more modestly "determines" richness, as in the title of the paper). With all the new analyses, particularly the rarefaction analysis, which gives equal weight to each taxon, there is no doubt that, given the data, species richness declines monotonically with elevation above the basal plain, as long as we disregard the taxonomic structure of the data. Of course, elevation itself does not mean anything, and temperature is linearly correlated with elevation, so the inference that temperature is strongly correlated with species richness is unassailable, for this dataset.

But does that mean that the temperature gradient *causes* the species richness gradient? I think that is seriously over-reaching, for several reasons—beyond the usual platitude the correlation is not causation. The most important point is that, as Fig. 1 plainly shows, temperature is poorly correlated with richness for many of the component taxa, so more heat equals more species is clearly not a universal correlation, not to mention a universal explanation. Why dismiss the rich diversity of patterns in favor of a statistical generality? Although this case is not a classic example of Simpson's paradox, the amalgamation of diverse data to infer general causality always has its perils. For a different approach to diversity of pattern, see my recently published paper on "midpoint attractors" in the latest issue Ecology Letters.

Second, although band area on the mountain itself indeed appears to explain little of the overall pattern, in the context of the other variables considered, I am haunted by the vision and potential influence of the gigantic area of the basal plain, which is not taken into account in the statistical analysis, for both contemporary and historical influences on richness on this very young mountain. For this reason, including MDE in the analysis makes little sense, as I said in my earlier review. In no sense is the base of the mountain is not a hard boundary. That others have "successfully" carried out MDE analyses in such contexts (including some of my collaborators) does not change my view on the matter for this case.

The rarefaction analysis adequately accounts for unequal contributions of different taxa to the overall pattern. But undersampling is not the same as unequal sampling, which is why I

suggested assessing statistical coverage for the different taxon surveys. The authors obligingly computed coverage for each group, and reported the raw results in Supplemental Table 1, but provide no summary or interpretation whatsoever for these data, not even in the table caption (in which they define “coverage” exactly backwards; the definition given is for 1 – coverage). Assuming the data in the table are actually coverage, values range from single digits (0.02) to 1.00. Is the reader supposed to analyze this table and somehow be reassured that undersampling did not affect the overall result? Are the sites in order of elevation? Does coverage vary with elevation?

In my first review, I wrote, “The notions of ‘sampling intensity’ and ‘completeness’ are never defined in this study.” That is still the case.

I have attached my markup of the draft, which some small corrections and suggestions. The red ovals highlight sentences of concern, for one reason or another.

Robert K. Colwell
University of Connecticut

Reviewer #2 (Remarks to the Author):

NCOMMS-16-00762A: Determinants of elevational biodiversity gradients change from single taxa to the multi-taxa community level

Peters et al.

For Nature Communications

I’ll start by repeating myself: what an excellent paper. It will surely become a classic in elevational gradient studies, be used in undergraduate lectures and graduate seminars, and likely appear in biogeography textbooks.

In my previous review, I tried to play ‘the skeptical reviewer’ as a way to help Peters et al. tell the best possible story. I can see that some of my comments were helpful and implemented. Others were not, but Peters et al. argued convincingly about why they did what they did. This is a much improved paper and ready for publication.

I don’t usually do this. In fact, I don’t think I ever have. But, as an unbiased observer, I’d like to say that Peters et al. have done a fabulous job in responding to the other reviewer’s comments

and criticisms.

Reviewers' comments:

Reviewer #1 (Remarks to the Author):

I commend the authors for undertaking extensive additional analyses, some quite original, in response to reviewers' comments, including my own, and for moderating over-interpretation of the results as support for the "kinetic energy hypothesis." The study is now better integrated with the existing literature. Overall, it is much strengthened, and certainly deserves to be published.

Response: Many thanks for critically reading the manuscript again and commenting on it.

But I still have concerns, especially about the interpretation and presentation of the main result: that temperature actually "drives" overall species richness of multiple clades (or slightly more modestly "determines" richness, as in the title of the paper). With all the new analyses, particularly the rarefaction analysis, which gives equal weight to each taxon, there is no doubt that, given the data, species richness declines monotonically with elevation above the basal plain, as long as we disregard the taxonomic structure of the data. Of course, elevation itself does not mean anything, and temperature is linearly correlated with elevation, so the inference that temperature is strongly correlated with species richness is unassailable, for this dataset.

But does that mean that the temperature gradient **causes** the species richness gradient? I think that is seriously over-reaching, for several reasons—beyond the usual platitude the correlation is not causation. The most important point is that, as Fig. 1 plainly shows, temperature is poorly correlated with richness for many of the component taxa, so more heat equals more species is clearly not a universal correlation, not to mention a universal explanation. Why dismiss the rich diversity of patterns in favor of a statistical generality? Although this case is not a classic example of Simpson's paradox, the amalgamation of diverse data to infer general causality always has its perils. For a different approach to diversity of pattern, see my recently published paper on "midpoint attractors" in the latest issue Ecology Letters.

Response: We addressed this comment by rewriting parts of the discussion. First, we were more careful in choosing words describing the detected statistical effects of temperature on species richness (e.g. we changed "...driver of species richness" to "...predictor of species richness"). Second, we added critical sentences pointing to the correlative nature of our study which does not provide true evidence that the variation in temperature is *causing* the variation in species richness (here a long-term experimental study would be needed). We also state that there are alternative methods (i.e. midpoint attractors) to the ones used to analyze elevational diversity data, in particular in combination with geometric constraints, which were not considered in our study. However, we do not share the position that (a) temperature is poorly correlated with richness for many component taxa, and (b) that a low correlation for component taxa speaks against the role of temperature as a driver of species richness. We therefore changed many but not all sentences that were marked by Prof. Colwell.

Concerning (a): running multiple regression models within a multi-model inference framework clearly shows that temperature is an important (statistical) predictor of species richness for most component taxa (Table1), with mostly strong, positive effects. Indeed, in terms of consistency across taxa, it is the only predictor with significant positive effects in a large number of taxa. However, for many component taxa temperature was not the only variable significantly affecting species richness. The parallel influence of other variables on species richness may explain at least in parts why species richness of component taxa did not exactly follow the elevation/temperature gradient. Concerning (b): Other authors already underscored that the phylogenetic/taxonomic coverage may be important for finding strong relationships between energy predictor variables and species richness (e.g. Hurlbert and Stegen 2014, Ecology Letters; Allen et al. 2002). We explained the reasoning for this in the introduction of the manuscript. As emphasized in the introduction and our previous response letter we are

convinced that the broad taxonomic coverage is a major strength of our study and allows testing the impact of environmental drivers on overall species richness. However, as we see that the topic has to be further developed in the ecological literature, we are currently preparing a conceptual manuscript on the importance of taxonomic/phylogenetic scale for inference on drivers of species richness. Concerning the relationship between temperature and species richness we developed in this manuscript a simulation model using the *tree.musse* function in the R package *diversitree* showing that under the assumption of some degree of niche conservatism, subclades may not show strong correlations of species richness and temperature even when temperature is set to be the only parameter influencing of species richness. Please find a figure with its legend from the manuscript below.

It was also not our aim to 'dismiss the rich diversity of patterns in favor of a statistical generality' and we think that we spent much of the manuscript space on the analyses of patterns and potential predictors of species richness in all component taxa. However, in addition to component taxa analyses, we point to a general pattern which is evident at the community level. Our data allows for the first time to compare the predictive strength of basic ecological hypotheses regarding elevational species richness patterns at this level of biodiversity.

[Redacted]

Figure 2. Phylogenetic inclusiveness and its effects on diversity-temperature relationships under conservative trait evolution. Shown is the result of a *tree.musse* simulation (R package *diversitree*) of pure temperature-dependent diversification with the following parameters: speciation rate 0.18 – 0.26 from coldest (blue) to warmest regions (red), extinction rate in all regions = 0.1 and conservative, gradual evolution (left; rate of lineage niche transition = 0.04 for adjacent temperatures). The simulations ran until 8000 extend lineages were generated. Under the assumption of low rates of niche transition in comparison to rates of diversification, the effect of temperature (x axis in small panels) on diversity (y axis) is evident in the most inclusive clade but not necessarily in smaller subclades.

Second, although band area on the mountain itself indeed appears to explain little of the overall pattern, in the context of the other variables considered, I am haunted by the vision and potential influence of the gigantic area of the basal plain, which is not taken into account in the statistical analysis, for both contemporary and historical influences on richness on this very young mountain. For this reason, including MDE in the analysis makes little sense, as I said in my earlier review. In no sense is the base of the mountain is not a hard boundary. That others have "successfully" carried out MDE analyses in such contexts (including some of my collaborators) does not change my view on the matter for this case.

Response: We addressed this comment by pointing out in the discussion section that the usefulness of the application of MDE as a predictor of species richness for mountains not extending to the sea level is a matter of debate and point to new developments in the analysis of MDE (Colwell et al. 2016).

The rarefaction analysis adequately accounts for unequal contributions of different taxa to the overall pattern. But undersampling is not the same as unequal sampling, which is why I suggested assessing statistical coverage for the different taxon surveys. The authors obligingly computed coverage for each group, and reported the raw results in Supplemental Table 1, but provide no summary or interpretation whatsoever for these data, not even in the table caption (in which they define "coverage" exactly backwards; the definition given is for $1 - \text{coverage}$). Assuming the data in the table are actually coverage, values range from single digits (0.02) to 1.00. Is the reader supposed to analyze this table and somehow be reassured that undersampling did not affect the overall result? Are the sites in order of elevation? Does coverage vary with elevation?

In my first review, I wrote, "The notions of 'sampling intensity' and 'completeness' are never defined in this study." That is still the case.

Response: Thanks for pointing to a mistake in the Supplementary Table 1 which we corrected. We now provide detailed interpretations of the Supplementary Table 1 (in which data on sampling coverage is presented) in the first section of the results. The terms sampling intensity, sampling completeness and sampling coverage were repeatedly used under similar contexts. We now rephrased some of the sentences to increase the precision of the text, explain the terms in the method section and the caption of Supplementary Table1, and refer to key papers from Chao and colleagues (in both the main text and in the Supplementary Information) in which the terms have been developed.

I have attached my markup of the draft, which some small corrections and suggestions. The red ovals highlight sentences of concern, for one reason or another.

Response: We checked all marked sentences and rephrased them.

Robert K. Colwell
University of Connecticut

Reviewer #2 (Remarks to the Author):

NCOMMS-16-00762A: Determinants of elevational biodiversity gradients change from single taxa to the multi-taxa community level

Peters et al.

For Nature Communications

I'll start by repeating myself: what an excellent paper. It will surely become a classic in elevational gradient studies, be used in undergraduate lectures and graduate seminars, and likely appear in biogeography textbooks.

In my previous review, I tried to play 'the skeptical reviewer' as a way to help Peters et al. tell the best possible story. I can see that some of my comments were helpful and implemented. Others were not, but Peters et al. argued convincingly about why they did what they did. This is a much improved paper and ready for publication.

I don't usually do this. In fact, I don't think I ever have. But, as an unbiased observer, I'd like to say that Peters et al. have done a fabulous job in responding to the other reviewer's comments and criticisms.

Response: Many thanks for these positive and encouraging comments!

** See Nature Research's author and referees' website at www.nature.com/authors for information about policies, services and author benefits

This email has been sent through the NPG Manuscript Tracking System NY-610A-NPG&MTS

Confidentiality Statement:

This e-mail is confidential and subject to copyright. Any unauthorised use or disclosure of its contents is

prohibited. If you have received this email in error please notify our Manuscript Tracking System Helpdesk team at <http://platformsupport.nature.com>.

Details of the confidentiality and pre-publicity policy may be found here <http://www.nature.com/authors/policies/confidentiality.html>

Privacy Policy | Update Profile

Reviewers' Comments:

Reviewer #1 (Remarks to the Author):

In their second revised version, I am pleased that the authors have made an effort to tone down the implication that their entirely correlative study confirms, rather than suggests, a causal role for temperature in the multi-clade pooling of species richness estimates in their data. In this regard, it is disappointing that the first word of the title (and the running head) remains “Determinants,” instead of the more appropriate term “Correlates.” In fact, the authors never once use the “determinants” to describe their own findings, in the entire body of the paper, so it does not seem appropriate in the title.

As I suspect the authors will agree, Rahbek (2005) (Ref. 38 in Peters et al., with the year mistakenly given as “2004”) is the definitive review, to date, of patterns of elevational richness (though there have been many more primary studies since 2005). The Mt. Kilimanjaro transect is, by necessity, “incomplete” or “shortened” in Rahbek’s classification, since it fails to cover elevations below 500 m asl.

In a pertinent passage, Rahbek wrote:

“Shortening the extent of the altitudinal gradient by omitting the lower end can result in unidirectional bias and the appearance of a continuous decrease in species richness independent of the actual differences in the true underlying pattern (Fig. 2C).”

In Lines 203-205 Peters et al. write: “A monotonous [sic] decrease of species richness with elevation, as found at the highest level of taxonomic coverage, appears to contrast with former meta-analyses^{37,38} finding overwhelmingly unimodal patterns of elevational diversity.” The appropriate frequency data in Rahbek (2005) is Panel D of his Figure 3 (“Shortened gradients, Standardized studies”), which shows that a very substantial proportion of studies, for shortened gradients, show monotonic (“monotonous” is not the correct word) declining patterns, just as Rahbek describes in the quotation above. “Overwhelming” is misleading. But more to the point, the authors fail to discuss the issue Rahbek raises. In fact, one has to read the Methods section of the present MS to learn that the transect begins above 700 m asl. I urge the authors to make appropriate revisions.

Robert K. Colwell

University of Connecticut

General response: Thanks for conducting another round of review of our manuscript and providing these additional helpful comments.

REVIEWERS' COMMENTS:

Reviewer #1 (Remarks to the Author):

In their second revised version, I am pleased that the authors have made an effort to tone down the implication that their entirely correlative study confirms, rather than suggests, a causal role for temperature in the multi-clade pooling of species richness estimates in their data. In this regard, it is disappointing that the first word of the title (and the running head) remains "Determinants," instead of the more appropriate term "Correlates." In fact, the authors never once use the "determinants" to describe their own findings, in the entire body of the paper, so it does not seem appropriate in the title.

Response: We changed in the title the term "Determinants" to "Predictors". We used the term "predictors" throughout the whole manuscript and think that it fits better than "correlates".

As I suspect the authors will agree, Rahbek (2005) (Ref. 38 in Peters et al., with the year mistakenly given as "2004") is the definitive review, to date, of patterns of elevational richness (though there have been many more primary studies since 2005). The Mt. Kilimanjaro transect is, by necessity, "incomplete" or "shortened" in Rahbek's classification, since it fails to cover elevations below 500 m asl.

In a pertinent passage, Rahbek wrote: "Shortening the extent of the altitudinal gradient by omitting the lower end can result in unidirectional bias and the appearance of a continuous decrease in species richness independent of the actual differences in the true underlying pattern (Fig. 2C)." In Lines 203-205 Peters et al. write: "A monotonous [sic] decrease of species richness with elevation, as found at the highest level of taxonomic coverage, appears to contrast with former meta-analyses^{37,38} finding overwhelmingly unimodal patterns of elevational diversity." The appropriate frequency data in Rahbek (2005) is Panel D of his Figure 3 ("Shortened gradients, Standardized studies"), which shows that a very substantial proportion of studies, for shortened gradients, show monotonic ("monotonous" is not the correct word) declining patterns, just as Rahbek describes in the quotation above. "Overwhelming" is misleading. But more to the point, the authors fail to discuss the issue Rahbek raises. In fact, one has to read the Methods section of the present MS to learn that the transect begins above 700 m asl. I urge the authors to make appropriate revisions.

Robert K. Colwell
University of Connecticut

Response: (1) We changed the publication year of Rahbek's study to 2005. Thanks for pointing to this error. (2) We changed "finding overwhelmingly unimodal patterns of elevational diversity" to "finding unimodal patterns of elevational diversity in a majority of studies". (3) We now added information on the extent of the studied elevational gradient in the introduction. Please also note that, additionally, the extent of the elevation gradient is shown in Figs. 1 and 3. (4) We added additional sentences in the discussion pointing to the findings of Rahbek (2005) that patterns of elevational diversity depend on the spatial extent of the studied elevational gradients and discuss it with respect to the elevational gradient of Mt. Kilimanjaro. We state that future studies on

mountains extending to the sea level and perhaps covering even more extreme environmental conditions at the upper end of the temperature gradient could provide interesting additional insights.